# Training-free Task Classification for Multi-Task Model Merging

## Abstract

The advent of foundation models, coupled with the pretraining-finetuning paradigm, has ignited a proliferation of various tasks and corresponding task-specific models. This has, in turn, spurred research into finding a unified system that can handle any input coming from various tasks, by combining models via weight interpolation. However, these methods assume the knowledge of which task distribution (or task ID) each input belongs to. While few recent works have attempted to design new merging methods that can handle scenarios where task ID is unknown (task-unknown scenarios), they require either additional training or multiple number of forward passes, undermining the efficiency of a unified framework. In this work, we aim to empower existing merging methods with the capability of handling task-unknown scenarios, without additional training or multiple number of forward passes. To this end, we reconceptualize the pursuit of model merging for task-unknown scenarios as a task-classification challenge: identifying the task distribution a given input data belong to. Leveraging Gaussian discriminant analysis (GDA), we introduce our method, MAD, which identifies the task identity of input data by comparing the **Ma**halanobis **D**istance between input features and each task-conditional Gaussian distribution. Consequently, MAD can be applied to existing model merging methods in an off-the-shelf manner to empower them with the capability to handle task-unknown scenarios. Experimental results demonstrate the effectiveness and flexibility of MAD for both computer vision and natural language processing domains, under task-unknown scenarios.

## 1 Introduction

With the emergence of foundation models (Radford et al., 2021; Achiam et al., 2023; Touvron et al., 2023), the pretraining-finetuning paradigm—the practice of pre-training foundation models followed by task-specific fine-tuning—has become one of the dominant approaches in many areas of machine learning (Brown et al., 2020; Radford et al., 2019; Wei et al., 2022; Wortsman et al., 2022). This pretraining-finetuning strategy leverages the knowledge acquired from massive datasets to achieve state-of-the-art performance on a wide range of downstream tasks, leading to a burst of task-specific models. However, the proliferation of these task-specific models presents new challenges related to knowledge and model management.

Accordingly, several works have attempted to integrate the knowledge from these task-specific models (or task experts) into a merged model via interpolation on task-experts weights (Ilharco et al., 2023; Matena & Raffel, 2022; Jin et al., 2023; Yadav et al., 2023; Wang et al., 2024; Gargiulo et al., 2024; Lu et al., 2024; Oh et al., 2025). In particular, several works have explored subspace-based model merging, which mitigates parameter interference by pruning unimportant parameters from individual task-specific models or a pre-merged model prior to merging (Wang et al., 2024; Huang et al., 2024; Gargiulo et al., 2024). However, these works require prior knowledge of the task distribution to which each new input belongs, thereby undermining their applicability to real-world scenarios.

In light of the challenge, few recent works have attempted to handle task-unknown scenario, where the task identity of a new input is unknown (Oh et al., 2025; Tang et al., 2024; Lu et al., 2024) by identify the task identity of each input and dynamically merging model parameters. While input-adaptive weight interpolation methods have led to substantial performance improvement without

| Method | Additional training | Forward passes | Task ID requirement |
|---|---|---|---|
| Subspace-based model merging | ✗ | $\mathcal{O}(1)$ | ✓ |
| *(Model merging methods for task-unknown scenarios)* | | | |
| TWIN-Merging | ✓ | $\mathcal{O}(1)$ | ✗ |
| DaWin | ✗ | $\mathcal{O}(N+T)$ | ✗ |
| **MAD(Ours)** | ✗ | $\mathcal{O}(1)$ | ✗ |

Table 1: **Model merging methods for task-unknown scenarios and their requirements.** Given $T$ task-specific models, $N$ denotes the number of test samples.

prior knowledge of the current task, they not only load all task-specific models into memory during inference but also require additional training (Lu et al., 2024; Tang et al., 2024) or multiple forward passes per task-specific model (Oh et al., 2025).

In this work, we introduce a plug-and-play method, MAD, that allows existing subspace-model merging methods to function independently of prior knowledge about the current evaluation task. We first note that the goal of finding input-adaptive task coefficients (or task routing) can be translated to classifying which task expert is an expert for a given new input data (or task classification). With the assumption that features of merged model can be modeled by task-conditional Gaussian distribution (Lee et al., 2018), we achieve task classification via formulating a generative (distance-based) classifier under Gaussian discriminant analysis (GDA). Specifically, the Mahalanobis distance between the input feature and task-conditional Gaussian distribution of each task is computed. In turn, task classification is performed using the computed distances to predict which task the current input belongs to. Then, for the predicted task, insignificant parameters are pruned, and the final weight interpolation is conducted. Consequently, the input is ultimately processed by the final merged model. Such GDA-based task classification enables subspace-based model merging without prior knowledge of the current evaluation task.

Experimental results across vision and NLP domains demonstrate that MAD substantially outperforms existing model merging methods for task-unknown scenarios that rely on additional training or multiple forward passes, despite that our method does not require additional training or multiple forward passes. Furthermore, MAD is based on efficient subspace-based model merging methods such as TALL-Mask (Wang et al., 2024) and EMR-Merging (Huang et al., 2024). This allows it to significantly reduce memory overhead by only requiring a single pre-merged parameters and task-specific binary pruning masks, rather than the full task-specific parameters for all tasks, as is typical of previous model merging methods for task-unknown scenarios methods. Table 1 explicitly details the comparison between MAD and other model merging methods for task-unknown scenarios. In conclusion, MAD effectively overcomes the limitations of both subspace-based model merging and model merging methods for task-unknown scenarios simultaneously.

## 2 RELATED WORK

The purpose of multi-task model merging is to build a multi-task model by combining the parameters of task-specific models. Task Arithmetic (Ilharco et al., 2023) merges models by task vectors, which represent the difference between the parameters of the task-specific models and the pre-trained model. This method assigns uniform task importance across tasks. These methods suffer from parameter interference because they merge without pruning insignificant parameters. Subsequently, subspace model merging approaches have emerged as a key approach to mitigate task interference by retaining only the most important parameters. TIES-Merging (Yadav et al., 2023) utilizes parameter pruning and sign election to generate task-specific masks to adjust the parameter importance. TALL-Mask (Wang et al., 2024) extracts significant task-specific parameters from a multi-task model based on the difference between the multi-task model and task-specific model parameters. EMR-Merging (Huang et al., 2024) simultaneously leverages a unified task vector that contains universal knowledge and masks and coefficients that contain task-specific knowledge for each task during merging. However, a crucial limitation of subspace model merging is its reliance on an explicit task ID during inference, which restricts its applicability. As a solution to limited adaptability, model merging for task-unknown scenarios has been introduced, an approach characterized

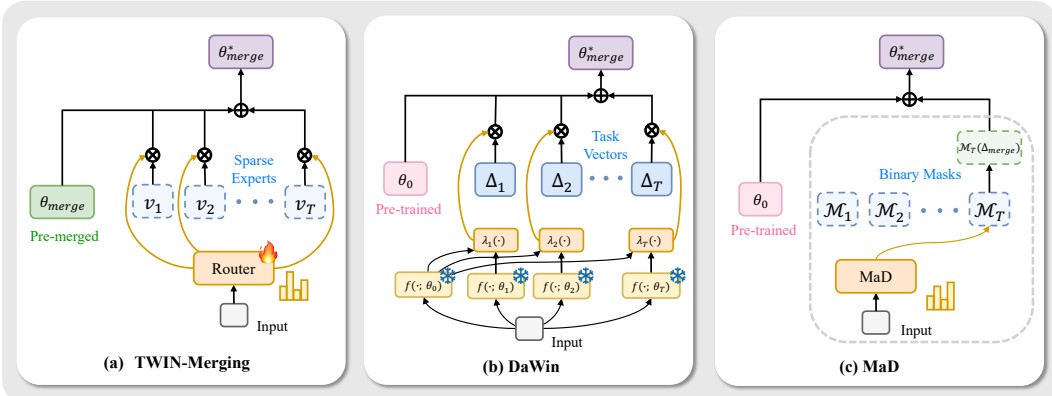

Figure 1: **Comparison of merging methods for task-unknown scenarios.** Subfigure (a) shows that TWIN-Merging performs inference with a single forward pass, but the coefficients are provided by a router that requires additional training. Subfigure (b) illustrates that DaWin, although training-free, computes coefficients by calculating entropy after multiple forward passes over each task-specific model based on the input. Subfigure (c) shows that MAD dynamically determines and applies binary pruning masks based on the input without requiring any additional training and with only one forward pass per sample.

by its adjustment of parameter importance based on test samples. TWIN-Merging (Lu et al., 2024) employs a router to calculate parameter importance. Based on this importance, it extracts task-specific experts, which are then combined with a shared expert possessing universal knowledge. This method incurs additional training costs from the router training. DaWin (Oh et al., 2025) calculates parameter importance using the Shannon entropy of the pre-trained model and task-specific models for the input sample and then merges the models using the calculated coefficients. Since this method requires forward passes from multiple models for entropy calculation, a key disadvantage is the linear increase in inference time. WEMoE (Tang et al., 2024) utilizes mixture-of-experts module to combine shared and task-specific experts per input, alleviating parameter interference, however at the cost of training both the router and additional experts and increasing the memory footprint. MoW-Merging (Ye et al., 2025) introduces a lightweight gating network trained on a small set of unlabeled samples to produce sample-wise task probabilities used as merging coefficients, enabling plug-and-play dynamic merging on top of static rules. However, it still relies on an auxiliary gating network and offline training on unlabeled data, in contrast to our fully training-free and router-free MAD

In contrast, our method eliminates the need for additional training and multiple forward passes for each input test sample, all while incurring low memory overhead. Figure 1 highlights the inherent efficiency of our proposed MAD when contrasted with model merging approaches for task-unknown scenarios.

## 3 BACKGROUND

**Problem setting.** Let $f : \mathcal{X} \times \Theta \to \mathcal{Y}$ be a pre-trained model with its parameters $\boldsymbol{\theta}_0 \in \Theta$. For every downstream task $t \in [1, T]$, under the pretraining-finetuning paradigm, $f$ is fine-tuned on the task-specific dataset $\mathcal{D}^{(t)} = \{(\boldsymbol{x}_i^{(t)}, y_i^{(t)})\}_{i=1}^{N_t}$, consisting of input samples $\boldsymbol{x}_i^{(t)} \in X^{(t)} \subseteq \mathcal{X}$ with the corresponding labels $y_i^{(t)} \in Y^{(t)} \subseteq \mathcal{Y}$. For a given input data $\boldsymbol{x}$, sampled from unknown task distribution $\tau$, the goal is to find an expertise (an expert model) that has the knowledge to handle the given input $\boldsymbol{x}$.

**Subspace-based model merging.** This approach mitigates parameter interference by pruning unnecessary parameters before merging (Yadav et al., 2023; Huang et al., 2024). These methods are grounded in the principle that neural networks are highly over-parameterized, enabling extensive parameter removal with a negligible impact on performance (Choudhary et al., 2020; He & Xiao, 2023; Marinó et al., 2023).

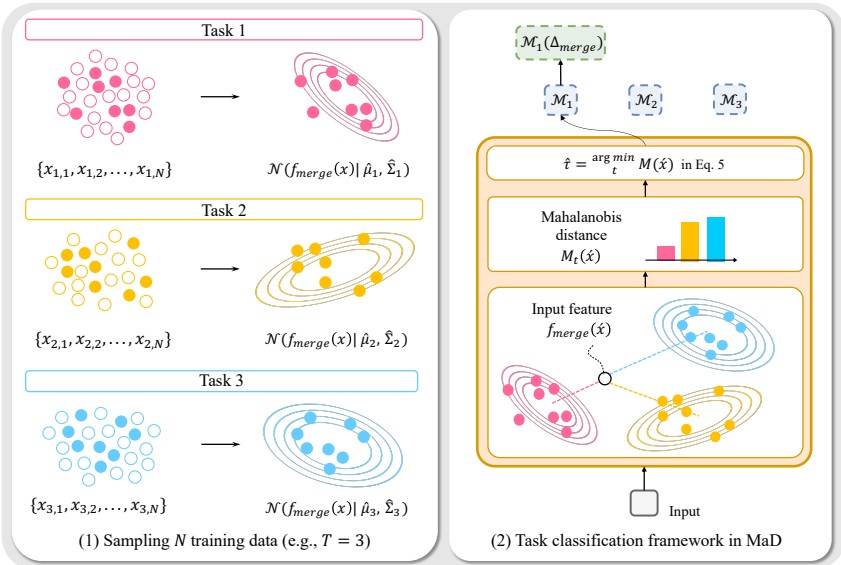

Figure 2: **Overview of MAD.** This figure provides a detailed illustration of the MAD framework, which outlines the internal processes within the gray dashed border box shown in Figure 1 (c). Our proposed method MAD performs task classification based on Mahalanobis distance, which then determines and selects the corresponding binary mask for the predicted task ID.

**Model merging for task-unknown scenarios.**   Arguing that the accurate prediction for each data sample $x$ requires varying model expertise, recent works have, instead, focused on model merging for task-unknown scenarios (Lu et al., 2024; Oh et al., 2025). Specifically, they aim to dynamically find model expertise (or interpolation coefficients) for each input data point:

$$\boldsymbol{\theta} = \sum_{t=1}^{T} \lambda_t(\boldsymbol{x})\boldsymbol{\theta}_t, \tag{1}$$

where the estimation of $\lambda_t(\boldsymbol{x})$ often requires the access to task-specific models (Lu et al., 2024; Oh et al., 2025). Therefore, unlike subspace-based model merging, model merging for task-unknown scenarios approaches do not require prior knowledge of task ID, making them more practical.

## 4   METHOD

### 4.1   MOTIVATION

**Limitations in previous model merging.**   Current subspace-based model merging methods mitigate parameter interference and enhance performance efficiently (Wang et al., 2024; Huang et al., 2024). By leveraging binary masks, these approaches achieve memory efficiency while effectively alleviating parameter interference. However, they require prior knowledge of the task ID being evaluated. Model merging for task-unknown scenarios, on the other hand, mitigates parameter interference by performing merging dynamically for each input (Lu et al., 2024; Oh et al., 2025). As these methods perform merging based on the input, they do not rely on prior knowledge of the task ID during inference. However, they incur significant memory overhead, as all expert parameters to be merged must be retained during inference.

In this work, we propose a novel method that combines the advantages of both subspace-based and model merging for task-unknown scenarios while mitigating their respective drawbacks. Specifically, our proposed method MAD empowers subspace-based model merging approaches with the ability to dynamically classify the task of the current input. Formally, we define task classification problem as follows:

**Definition 1** (Task classification problem). *We consider the task classification problem, where the goal is to determine the underlying task identity $\tau \in \{1, 2, \dots, T\}$ of a given input sample $\boldsymbol{x} \in \mathcal{X}$. Each task $t \in \{1, 2, \dots, T\}$ corresponds to a distinct data distribution $\mathcal{D}^{(t)}$ over the input space.*

*Let $\mathcal{X}$ denote the input space and $\mathcal{T} = \{1, 2, \ldots, T\}$ the set of task indices. Given an unlabeled sample $\boldsymbol{x} \sim \mathcal{D}^{(\tau)}$ from an unknown task $\tau \in \mathcal{T}$, the objective is to learn a classifier $\mathcal{R} : \mathcal{X} \to \mathcal{T}$ that predicts the correct task label:*

$$\hat{\tau} = \mathcal{R}(\boldsymbol{x}) = \arg\max_{t \in \mathcal{T}} s_t(\boldsymbol{x}),$$

*where $s_t(\boldsymbol{x})$ denotes a task-specific score or likelihood indicating how well the input $\boldsymbol{x}$ aligns with task $t$.*

This enables subspace-based model merging methods to perform merging without relying on prior knowledge of the task ID. Furthermore, because subspace-based model merging leverages binary masks instead of actual expert parameters to capture model expertise for each task, it exhibits lower memory overhead compared to previous merging methods for task-unknown scenarios.

**Why Mahalanobis distance for task classification?: Gaussian discriminant analysis (GDA).** We are motivated by previous works on classification that utilize Mahalanobis distance for various domains (Lee et al., 2018; Xu et al., 2020; Ren et al., 2021; Shazeer et al., 2017; Guo et al., 2018; Paeedeh et al., 2024). In particular, (Lee et al., 2018) have demonstrated that each feature distribution from deep neural networks can be well approximated by Gaussian distribution. Building upon this, we observed in Figure 3 that features corresponding to each task, extracted from the merged model, are well-separated. This observation demonstrates the feasibility of Gaussian discriminant analysis (GDA) for task classification.

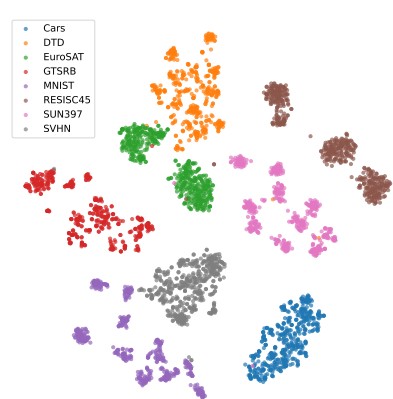

Figure 3: **Task-specific feature distributions from the merged model, approximated by Gaussian.**

Therefore, within GDA, we assume that the task-conditional likelihood $p(\boldsymbol{x}|\tau = t)$ follows a multivariate Gaussian distribution $\mathcal{N}(\boldsymbol{x}|\boldsymbol{\mu}_t, \boldsymbol{\Sigma}_t)$, where $\boldsymbol{\mu}_t$ and $\boldsymbol{\Sigma}_t$ are its mean vector and covariance matrix for task $t$, respectively. From this perspective, calculating the Mahalanobis distance between an input $\boldsymbol{x}$ and task distribution for task $t$ directly corresponds to evaluating the log-likelihood $\log p(\boldsymbol{x}|\tau = t)$ using estimated Gaussian parameters $\hat{\boldsymbol{\mu}}_t$ and $\hat{\boldsymbol{\Sigma}}_t$. Specifically, the log-likelihood is dominated by the term $-\frac{1}{2}(\boldsymbol{x} - \hat{\boldsymbol{\mu}}_t)^\top \hat{\boldsymbol{\Sigma}}_t^{-1}(\boldsymbol{x} - \hat{\boldsymbol{\mu}}_t)$, which is equivalent to the negative squared Mahalanobis distance (up to a constant). Therefore, minimizing the Mahalanobis distance is equivalent to maximizing the log-likelihood in the GDA classification rule $\hat{\tau}(\boldsymbol{x}) = \arg\max_t \log p(\tau = t) + \log p(\boldsymbol{x}|\tau = t)$. Under uniform task priors ($p(\tau = t)$ is constant), this simplifies to $\arg\min_t (\boldsymbol{x} - \hat{\boldsymbol{\mu}}_t)^\top \hat{\boldsymbol{\Sigma}}_t^{-1}(\boldsymbol{x} - \hat{\boldsymbol{\mu}}_t)$. This provides theoretical justification for the use of Mahalanobis distance in task classification. We further analyze why the use of Mahalanobis distance is well-suited for task classification, with theoretical support delineated in Appendix D.

This Mahalanobis-distance-based task classification demonstrates strong performance even with limited data (Lee et al., 2018) and obviates the need for additional training. Consequently, it offers good scalability when tasks are added or removed from the merging process.

## 4.2 ESTIMATING FEATURE DISTRIBUTION FOR EACH TASK

Before inference, we first extract feature distribution of merged model for each task. Each feature distribution is approximated as Gaussian distribution $\mathcal{N}(f_{\text{merge}}(\boldsymbol{x})|\hat{\boldsymbol{\mu}}_t, \hat{\boldsymbol{\Sigma}}_t)$, which is estimated with $N$ randomly gathered samples $\mathcal{D}_{\text{train}}^{(t)} = \{\bar{\boldsymbol{x}}_t^{(1)}, \bar{\boldsymbol{x}}_t^{(2)}, \ldots, \bar{\boldsymbol{x}}_t^{(N)}\} \subset \mathcal{X}$ of the training set of task $t$:

$$\hat{\boldsymbol{\mu}}_t = \frac{1}{N} \sum_{i=1}^{N} f_{\text{merge}}(\bar{\boldsymbol{x}}_t^{(i)}) \in \mathbb{R}^D, \tag{2}$$

$$\hat{\boldsymbol{\Sigma}}_t = \frac{1}{N-1} \sum_{i=1}^{N} (f_{\text{merge}}(\bar{\boldsymbol{x}}_t^{(i)}) - \hat{\boldsymbol{\mu}}_t)(f_{\text{merge}}(\bar{\boldsymbol{x}}_t^{(i)}) - \hat{\boldsymbol{\mu}}_t)^\top \in \mathbb{R}^{D \times D}, \tag{3}$$

where $f_{\text{merge}}(\bar{\boldsymbol{x}}_t) = f(\bar{\boldsymbol{x}}_t; \boldsymbol{\theta}_{\text{merge}})$ denotes the features output by the merged model and $D$ is the feature dimension. We employ weight averaging as the default merging method for feature distribution estimation and task classification: $\boldsymbol{\theta}_{\text{merge}} = \frac{1}{T} \sum_{t=1}^{T} \boldsymbol{\theta}_t$, where $\boldsymbol{\theta}_t$ are the expert parameters for task $t$ and $T$ is the number of tasks. For task $t$, $\hat{\boldsymbol{\mu}}_t$ and $\hat{\boldsymbol{\Sigma}}_t$ represent the empirical mean and covariance of features of the merged model, respectively. Unless otherwise indicated, a sample size of $N = 64$ is used by default for these estimations. To prevent the covariance matrix $\hat{\boldsymbol{\Sigma}}_t$ from becoming singular, which would preclude the calculation of Mahalanobis distance when computed directly as in Eq. 3, we add a small default value $\epsilon = 10^{-4}$ to its diagonal.

### 4.3 INFERENCE

**Task classification.** To predict the task ID of input test samples, we first compute the Mahalanobis distance $M(\acute{\boldsymbol{x}}) \in \mathbb{R}^{B \times T}$, where $B$ is the batch size and $T$ is the number of tasks. This distance is calculated between the features $f_{\text{merge}}(\acute{\boldsymbol{x}}) \in \mathbb{R}^{B \times D}$ of unknown batch test samples $\acute{\boldsymbol{x}} \in \mathcal{D}_{\text{test}} \subset \mathcal{X}$ and the empirical mean $\hat{\boldsymbol{\mu}}_t$ of each task $t$, using its corresponding empirical covariance $\hat{\boldsymbol{\Sigma}}_t$:

$$M_t^{(i)}(\acute{\boldsymbol{x}}) = \sqrt{(f_{\text{merge}}(\acute{\boldsymbol{x}}^{(i)}) - \hat{\boldsymbol{\mu}}_t)^\top \hat{\boldsymbol{\Sigma}}_t^{-1} (f_{\text{merge}}(\acute{\boldsymbol{x}}^{(i)}) - \hat{\boldsymbol{\mu}}_t)}, \tag{4}$$

where $i \in \{1, 2, \dots, B\}$. This distance serves as a predicted task-specific logit for task classification. The logits $M(\acute{\boldsymbol{x}})$ are then utilized to predict the task IDs $\hat{\boldsymbol{\tau}}$ of batch input samples as follows:

$$\hat{\boldsymbol{\tau}} = \arg\min_t M(\acute{\boldsymbol{x}}). \tag{5}$$

Intuitively, for each sample $\acute{\boldsymbol{x}}^{(i)}$ within the batch, the task with the smallest distance to that sample is predicted as its associated task.

**Group batch inference.** After task classification, for each predicted task ID $\hat{\tau} \in \hat{\boldsymbol{\tau}}$, we select the corresponding samples $\acute{\boldsymbol{x}}_{\hat{\tau}} \subset \acute{\boldsymbol{x}}$ and the binary mask $\mathcal{M}_{\hat{\tau}} \in \{\mathcal{M}_1, \mathcal{M}_2, \dots \mathcal{M}_T\}$ capable of extracting model expertise from the previously-calculated merged (pre-merged) task vector $\boldsymbol{\Delta}_{\text{merge}}$. Finally, merging is performed using this binary mask, and inference is conducted on the selected samples with the resulting merged parameters $\boldsymbol{\theta}_{\text{merge}}^\star = \boldsymbol{\theta}_0 + \mathcal{M}_{\hat{\tau}}(\boldsymbol{\Delta}_{\text{merge}})$:

$$\hat{\boldsymbol{y}}_{\hat{\tau}} = \arg\max_c f(\acute{\boldsymbol{x}}_{\hat{\tau}}; \boldsymbol{\theta}_{\text{merge}}^\star) \subset \hat{\boldsymbol{y}}, \tag{6}$$

where $c$ denotes the class ID of the current evaluation task, $\hat{\boldsymbol{y}}$ represents the overall batch predictions, and $\hat{\boldsymbol{y}}_{\hat{\tau}}$ specifically refers to the predictions for the selected task within the batch from $\hat{\boldsymbol{y}}$.

We apply task classification framework of MAD to the below subspace-based model merging methods:

- TALL-Mask + Task Arithmetic (TM-TA) (Ilharco et al., 2023): This method first computes a merged task vector $\boldsymbol{\Delta}_{\text{merge}} = \alpha \sum_{t=1}^{T} \boldsymbol{\Delta}_t$ using Task Arithmetic (Wang et al., 2024). Here, $\boldsymbol{\Delta}_t = \boldsymbol{\theta}_t - \boldsymbol{\theta}_0$ is a task vector encapsulating the model expertise for task $t$, where $\boldsymbol{\theta}_t$ are the parameters of the expert model for task $t$, and $\boldsymbol{\theta}_0$ is the pre-trained parameters. The hyperparameter $\alpha$ scales these task vectors. Subsequently, for each task $t$, a binary mask $\mathcal{M}_t$ is calculated to extract model expertise from the merged task vector $\boldsymbol{\Delta}_{\text{merge}}$. This calculation is based on the magnitude of the difference between the merged task vector $\boldsymbol{\Delta}_{\text{merge}}$ and the expert task vector $\boldsymbol{\Delta}_t$: $\mathcal{M}_t = \mathbb{1}\{|\boldsymbol{\Delta}_t| \geq |\boldsymbol{\Delta}_{\text{merge}} - \boldsymbol{\Delta}_t| \cdot \lambda_t\}$. The hyperparameter $\lambda_t$ determines how much information to extract from the merged task vector. Finally, the calculated binary mask $\mathcal{M}_t$ for task $t$ is used to compute the final merged parameters $\mathcal{M}_t(\boldsymbol{\Delta}_{\text{merge}}) = \mathcal{M}_t \odot \boldsymbol{\Delta}_{\text{merge}}$, where $\odot$ denotes the element-wise product operator.
- TALL-Mask + TIES-Merging (TM-TIES) (Wang et al., 2024): This method is one of the variants of TALL-Mask. Instead of computing the merged task vector $\boldsymbol{\Delta}_{\text{merge}}$ using Task Arithmetic, this variant leverages TIES-Merging for its calculation (Yadav et al., 2023). The remaining steps are identical to those previously explained.
- EMR-Merging (EMR) (Huang et al., 2024): This method first constructs a merged task vector $\boldsymbol{\Delta}_{\text{merge}}$ by applying a sign-and-magnitude-based aggregation to individual task vectors. For each task $t$, a corresponding mask $\mathcal{M}_t = \mathbb{1}\{\boldsymbol{\Delta}_t \odot \boldsymbol{\Delta}_{\text{merge}} > 0\}$ and rescaler

Table 2: **Multi-task performance of input-dependent merged CLIP ViT models for computer vision tasks across different number of tasks**. We report experimental results for ViT-B/32, ViT-B/16, and ViT-L/14 on 8, 14, and 20 vision tasks. Bold values represent the best performance among all methods, excluding the individual task-specific baselines.

| Method | ViT-B/32 | | | ViT-B/16 | | | ViT-L/14 | | |
|---|---|---|---|---|---|---|---|---|---|
| | 8 tasks | 14 tasks | 20 tasks | 8 tasks | 14 tasks | 20 tasks | 8 tasks | 14 tasks | 20 tasks |
| Fine-tuned | 92.8 | 90.9 | 91.3 | 94.7 | 92.8 | 92.8 | 95.9 | 94.3 | 94.8 |
| Weight Averaging | 66.3 | 64.3 | 61.0 | 72.2 | 69.5 | 65.3 | 79.6 | 76.7 | 71.6 |
| Task Arithmetic (Task-known) | 70.8 | 65.3 | 60.5 | 75.4 | 70.5 | 65.8 | 84.9 | 79.4 | 74.0 |
| TIES-Merging (Task-known) | 75.1 | 68.0 | 63.4 | 79.7 | 73.2 | 68.2 | 86.9 | 79.5 | 75.7 |
| *(Model merging methods for task-unknown scenarios)* | | | | | | | | | |
| Task Arithmetic (Task-unknown) | 61.3 | 38.2 | 23.6 | 68.8 | 45.5 | 25.6 | 83.4 | 66.6 | 42.7 |
| TIES-Merging (Task-unknown) | 74.0 | 65.0 | 58.2 | 79.6 | 71.3 | 65.8 | 86.9 | 78.7 | 74.4 |
| TWIN-Merging | 84.0 | 70.0 | 57.5 | 91.4 | 78.4 | 63.1 | 93.7 | 86.2 | 74.8 |
| DaWin | 89.0 | 73.8 | 52.8 | 87.1 | 77.8 | 62.8 | 91.6 | 82.6 | 77.5 |
| WEMoE | 90.4 | 83.1 | 74.4 | 93.1 | 84.0 | 76.6 | 94.8 | 87.0 | 75.7 |
| MoW-Merging | 88.1 | 83.2 | 79.3 | 93.7 | 79.3 | 78.2 | 94.9 | 78.8 | 81.8 |
| **EMR + MAD (Ours)** | 90.4 | 86.8 | 85.6 | 92.8 | 89.8 | 88.7 | **95.0** | **92.5** | **92.0** |
| **TM-TA + MAD (Ours)** | **92.0** | **89.4** | **88.7** | **93.9** | 91.0 | **91.1** | 92.1 | 89.0 | 89.8 |
| **TM-TIES + MAD (Ours)** | 91.9 | 89.1 | 88.2 | 93.8 | **91.4** | 90.7 | 93.9 | 90.3 | 90.6 |

$\delta_t$ are then computed. This rescaler is for aligning the parameter magnitude of the pre-merged model with the expert model. The final merged parameters $\theta^\star_{\text{merge}}$ are finally derived by applying these task-specific masks and rescalers to the pre-merged task vector $\boldsymbol{\Delta}_{\text{merge}}$:

$$\mathcal{M}_t(\boldsymbol{\Delta}_{\text{merge}}) = \delta_t \cdot \mathcal{M}_t \odot \boldsymbol{\Delta}_{\text{merge}}.$$

In conclusion, MAD empowers subspace-based model merging methods with the ability to perform merging for task-unknown scenarios based on input, even without prior knowledge of task IDs. Furthermore, by utilizing only a single pre-merged model and binary masks, it enables memory-efficient inference. A detailed description of the complete MAD procedure is provided in Algorithm 1.

# 5 EXPERIMENTS

## 5.1 SETUP

**Baselines.** We compared applying our proposed method to existing subspace-based model merging methods against several baselines. These include individual task-specific models and established subspace model merging techniques such as TALL-Mask + Task Arithmetic, TALL-Mask + TIES (Wang et al., 2024; Ilharco et al., 2023; Yadav et al., 2023), and EMR-Merging (Huang et al., 2024). Furthermore, we compare against other input-dependent model merging methods for comparison: DaWin (Oh et al., 2025), TWIN-Merging (Lu et al., 2024), WEMoE (Tang et al., 2024) and MoW-Merging (Ye et al., 2025). For MoW-Merging (Ye et al., 2025), we have adopted its best-performing configuration as reported in the original paper: AdaMerging++ w/ MoW-Merging. This setup utilized TrivialAugment for data augmentation, specifically applied during the training of its gating network, and set $m = 10$, where $m$ denotes the total number of attention and MLP layers to which method is applied.

To facilitate a fair comparison in task-unknown scenarios, we have adapted Task Arithmetic (Ilharco et al., 2023) and TIES-Merging (Yadav et al., 2023) by adopting a fixed parameter setting across all tasks. For Task Arithmetic (Ilharco et al., 2023), we set the scaling coefficient $\lambda = 0.4$. For TIES-Merging (Yadav et al., 2023), we used $K = 20$ and $\lambda = 1$, consistent with configurations suggested in its paper (Yadav et al., 2023).

Table 3: **Multi-task performance of merged T5-large across 7 NLP tasks.**. Bold values represent the best performance among all methods, excluding the individual task-specific baselines. Underlines indicate the case where fine-tuning performance is surpassed.

| Method | PAWS | QASC | QuaRTz | StoryCloze | WikiQA | Winogrande | WSC | Avg. |
|---|---|---|---|---|---|---|---|---|
| Fine-tuned | 94.4 | 98.9 | 87.8 | 90.8 | 96.0 | 74.7 | 79.2 | 88.8 |
| Weight Averaging | 61.3 | 82.6 | 70.5 | 53.7 | 63.2 | 49.7 | 36.1 | 59.6 |
| Task Arithmetic | 77.8 | 96.0 | 78.6 | 86.4 | 59.1 | 62.3 | 52.8 | 73.3 |
| TIES-Merging | 81.5 | 96.2 | 80.1 | 83.6 | 64.9 | 66.5 | 65.3 | 76.9 |
| *(Model merging methods for task-unknown scenarios)* | | | | | | | | |
| TWIN-Merging | 93.2 | 96.0 | **87.1** | 85.6 | 77.1 | 70.8 | **69.7** | 82.8 |
| DaWin | 85.6 | 96.1 | 81.4 | 73.2 | 70.2 | 68.7 | 57.9 | 76.2 |
| **EMR + MAD** | 95.4 | 96.4 | 83.0 | **89.7** | 93.8 | **95.0** | 59.0 | **87.5** |
| **TM-TA + MAD** | **96.2** | 96.6 | 84.5 | 86.6 | 93.2 | 94.5 | 41.0 | 84.7 |
| **TM-TIES + MAD** | 95.6 | **97.0** | 83.5 | 89.5 | **94.3** | 94.2 | 42.0 | 85.2 |

**Evaluation settings.** For evaluation, we follow the settings of TALL-Mask (Wang et al., 2024) for the computer vision tasks, and the settings of TIES-Merging (Yadav et al., 2023) for the 7 NLP tasks. We evaluate our method across four task scenarios. The 8 computer vision task scenario consists of: (1) SUN397 (Xiao et al., 2016), (2) Cars (Krause et al., 2013), (3) RESISC45 (Cheng et al., 2017), (4) EuroSAT (Helber et al., 2019), (5) SVHN (Netzer et al., 2011), (6) GTSRB (Stallkamp et al., 2011), (7) MNIST (Deng, 2012), and (8) DTD (Cimpoi et al., 2014). The 14 computer vision task scenario adds: (9) CIFAR100 (Krizhevsky et al., 2009), (10) STL10 (Coates et al., 2011), (11) Flowers102 (Nilsback & Zisserman, 2008), (12) OxfordIIITPet (Parkhi et al., 2012), (13) PCAM (Veeling et al., 2018), and (14) FER2013 (Goodfellow et al., 2013). The 20 computer vision task scenario further includes: (15) EMNIST (Cohen et al., 2017), (16) CIFAR10 (Krizhevsky et al., 2009), (17) Food101 (Bossard et al., 2014), (18) FashionMNIST (Xiao et al., 2017), (19) RenderedSST2 (Socher et al., 2013; Radford et al., 2019), and (20) KMNIST (Clanuwat et al., 2018). The 7 NLP task scenario comprises: (1) QASC (Khot et al., 2020), (2) WikiQA (Yang et al., 2015), (3) QuaRTz (Tafjord et al., 2019) for question answering, (4) PAWS (Zhang et al., 2019) for paraphrase identification, (5) Story Cloze (Sharma et al., 2018) for sentence completion, (6) Winogrande (Sakaguchi et al., 2020), (7) WSC (Levesque et al., 2012) for coreference resolution. We perform merging on CLIP (Radford et al., 2021) ViT-{B/32, B/16, L/14} (Dosovitskiy et al., 2021) for the computer vision tasks and on T5-large (Raffel et al., 2020) for the NLP tasks. The model checkpoints used in our experiments are obtained from the following sources: TALL-Mask (Wang et al., 2024) (for the computer vision tasks) and TIES-Merging (Yadav et al., 2023) (for the 7 NLP tasks).

## 5.2 MAIN RESULTS

**Computer vision tasks.** Experimental results for merging ViT-B/32, ViT-B/16 and ViT-L/14 across the 8, 14, and 20 vision tasks are summarized in Table 2. These results demonstrate that our approach, by enabling model merging for task-unknown scenarios when integrated with subspace-based techniques, consistently surpasses existing model merging methods for task-unknown scenarios across various model sizes, highlighting the significance of our proposed methodology. We also investigate the scalability of our method by evaluating its performance on task sets comprising more than 14 tasks. Our method consistently maintains the performance of existing subspace-based model merging techniques. Beyond merely preserving performance, our method demonstrates robust and accurate task classification capabilities, even with an expanded number of tasks.

**NLP tasks.** We further assess the effectiveness of our method by evaluating its performance on NLP tasks, in addition to the vision tasks. The corresponding experimental results for the NLP tasks are shown in Table 3. Our method demonstrates near fine-tuning performance even on NLP tasks. Notably, it surpasses fine-tuning performance on PAWS and Winogrande. This suggests that the Mahalanobis distance is capable of identifying a superior model compared to the original expert model.

Table 4: **Inference cost with CLIP ViT-B/32.** We report the computation costs of model merging methods for task-unknown scenarios across all tasks in the 8 computer vision task scenario.

| Method | Batch inference | Inference cost (per input) | VRAM (GB) | Avg. performance |
|---|---|---|---|---|
| *(Model merging methods for task-unknown scenarios)* | | | | |
| Task-Arithmetic (Task-unknown) | ✓ | 0.0008s | 1.0 | 61.3 |
| TIES-Merging (Task-unknown) | ✓ | 0.0008s | 1.0 | 74.0 |
| TWIN-Merging | ✗ | 0.03s | 5.6 | 84.0 |
| DaWin w/o mixture modeling | ✗ | 0.63s | 5.5 | 84.2 |
| DaWin | ✓ | 0.001s | 9.2 | 89.0 |
| WEMoE | ✗ | 0.02s | 4.3 | 90.4 |
| MoW-Merging | ✓ | 0.008s | 4.9 | 88.1 |
| **EMR + MAD (Ours)** | ✓ | 0.002s | 1.7 | 90.4 |
| **TM-TA + MAD (Ours)** | ✓ | 0.004s | 1.1 | 92.0 |
| **TM-TIES + MAD (Ours)** | ✓ | 0.004s | 1.1 | 91.9 |

Table 5: **Task classification accuracy (%) on seen versus unseen tasks under distribution shift.** The "seen" split corresponds to the 8-task vision benchmark used in the main experiments, whereas the "unseen" split consists of the additional 6 datasets from the 14-task benchmark that are not included in the 8-task set. Averaged performance across all 14 tasks is reported in the last column.

| Method | Seen (8 tasks) | Unseen (6 tasks) | Overall (14 tasks) |
|---|---|---|---|
| Task Arithmetic (Task-unknown) | 61.3 | 44.3 | 54.0 |
| TIES-Merging (Task-unknown) | 74.0 | 55.2 | 65.9 |
| DaWin | 89.0 | 57.5 | 75.5 |
| TWIN-Merging | 83.4 | 62.6 | 74.5 |
| **EMR+MAD (Ours)** | 90.5 | 57.8 | 76.5 |
| **EMR+TM-TA (Ours)** | **92.1** | 63.8 | 78.0 |
| **EMR+TM-TIES (Ours)** | 91.9 | **64.4** | **80.1** |

**Robustness to unseen tasks and feature shift.** To assess how robust our approach is to distributional shift when new tasks are introduced, we further evaluate task classification in a setting with both seen and unseen tasks. Concretely, we treat the 8-task vision benchmark used in the main experiments as the set of *seen* tasks, and use the additional 6 datasets from the 14-task benchmark as *unseen* tasks. As summarized in Table 5, classical subspace-based merging methods such as Task Arithmetic and TIES-Merging, which require known task IDs at inference time, perform reasonably on seen tasks but suffer substantially on unseen tasks. Among task-unknown baselines, DaWin and TWIN-Merging improve robustness to unseen tasks, but still exhibit a noticeable drop in accuracy compared to seen tasks. In contrast, MAD obtain the best overall performance. These results indicate that incorporating the MAD enables subspace-based merging methods to handle unseen tasks effectively, thereby mitigating the impact of feature distribution shift as the task set evolves.

## 5.3 COMPUTATIONAL COSTS

In this section, we investigate the efficiency of our method, particularly in comparison to another input-dependent model merging method: TWIN-Merging (Lu et al., 2024), DaWin (Oh et al., 2025), WEMoE (Tang et al., 2024) and MoW-Merging (Ye et al., 2025). As shown in Table 4, MAD demonstrates substantially faster inference, significantly higher memory efficiency, and superior overall performance compared to other dynamic approaches. Specifically, efficiency of MAD is highlighted by inference times that are almost 10 times faster than TWIN-Merging. Our approach facilitates group batch processing for samples belonging to the same task ID. This capability allows for significantly faster inference speeds compared to traditional sample-wise methods.

As for the memory costs, our method drastically reduces memory requirements. Existing merging methods for task-unknown scenarios face substantial memory challenges because they necessitates storing a complete task-specific model for every potential task. This results in a linear increase in memory cost with the number of merged tasks. Our method, however, combines with a subspace-based architecture, consequently requiring storage of only the merged task vector and task-specific masks, rather than full models. This design enables a remarkable reduction in memory

cost, achieving a decrease of more than 5 times compared to existing model merging approaches for task-unknown scenarios, all while simultaneously delivering the highest performance. Furthermore, our method is over 8 times more memory-efficient than DaWin, which also utilizes batch-wise processing, while maintaining comparable processing speeds.

## 5.4 TASK CLASSIFICATION

We have conducted experiments to validate the effectiveness of our training-free Mahalanobis distance approach for task classification, as shown in Table 6. Our analysis demonstrates that our method achieves superior classification performance, eliminating the need for any additional training. Furthermore, our method, even with only 64 samples per task, yields performance similar to that achieved us-

Table 6: **Task classification performance with CLIP ViT-B/32**

| Method | 8 tasks | 14 tasks | 20 tasks |
|---|---|---|---|
| MAD($N = 4$) | 90.5 | 89.9 | 80.5 |
| MAD($N = 8$) | 93.5 | 91.9 | 87.4 |
| MAD($N = N_{\text{train}}$) | 98.6 | 97.5 | 96.2 |
| MAD (our final model: $N = 64$) | 98.0 | 96.9 | 94.1 |

ing the entire training dataset. Here, $N_{\text{train}}$ represents the total number of training samples. This advantage is consistently maintained across diverse task scales, including experiments with 8, 14, and 20 tasks, clearly demonstrating its robustness and broad applicability.

## 5.5 ABLATION STUDY

Table 7: **Ablation studies.** We conduct an ablation study examining the impact on task classification performance: using Euclidean distance instead of Mahalanobis distance (second row), using Cosine similarity instead of Mahalanobis distance (third row), using only the diagonal values of the task-specific Gaussian distribution's covariance (fourth row), and our final model (fifth row).

| Method | Cars | DTD | EuroSAT | GTSRB | MNIST | RESISC45 | SUN397 | SVHN | Avg. |
|---|---|---|---|---|---|---|---|---|---|
| Euclidean dist. | 99.8 | 97.0 | 96.0 | 97.3 | 99.9 | 89.7 | 97.6 | 96.5 | 96.7 |
| Cosine sim. | 99.9 | 96.9 | 97.3 | 96.7 | 99.9 | 88.2 | 95.7 | 97.3 | 96.5 |
| Diagonal cov. | 98.3 | 76.5 | 54.1 | 95.3 | 98.5 | 82.5 | 99.9 | 78.8 | 85.5 |
| **MAD** | 99.9 | 98.5 | 99.9 | 98.8 | 99.9 | 96.6 | 92.1 | 98.6 | 98.0 |

To understand the influence of different design choices on the performance of MAD, we have conducted an ablation study. We investigate the impact of incorporating task-specific covariance in the task classification, and the effectiveness of different routing mechanisms. This study is based on the ViT-B/32 model and the 8 vision task scenario. Table 7 summarizes the findings of this ablation study.

An analysis of Table 7 clearly shows that considering task-specific feature distributions through the use of Mahalanobis distance for task classification is crucial for achieving optimal performance. Specifically, our full Mahalanobis distance approach significantly outperforms using only the diagonal elements of the covariance matrix, underscoring the importance of capturing full covariance structures. Furthermore, it consistently yields superior performance to both Euclidean distance and cosine similarity-based approaches. This indicates that effectively accounting for the underlying data distribution, particularly its complete variance and covariance structure, is beneficial for accurate task classification. Therefore, accounting for the full data distribution is crucial, as it leads to more effective and robust task classification compared to using simpler distance metrics.

## 6 CONCLUSION

In this work, we introduce MAD, a novel framework that empowers existing subspace-based model merging methods to operate effectively under task-unknown scenarios. By reformulating the merging process for task-unknown scenarios as a task classification problem and leveraging Gaussian Discriminant Analysis with Mahalanobis distance, our method is able to identify the task ID of each input without requiring additional training or multiple forward passes. Our method extends the applicability of existing subspace-based model merging methods to real-world settings where task information is unavailable. Our experiments demonstrate that our approach outperforms existing model merging methods for task-unknown scenarios across vision and NLP domains.

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

## A  TABLE OF CONTENTS

We provide the following items in this Appendix:

## B  PSEUDOCODE OF MAD

---

**Algorithm 1** MAD

---

**Require:** Pre-merged task vector $\boldsymbol{\Delta}_{\mathrm{merge}}$, binary masks $\{\mathcal{M}_t\}_{t=1}^{T}$ training samples $\{\bar{\boldsymbol{x}}_t^{(i)}\}_{i=1}^{N}$ per task $t$, small constant $\epsilon$ (for covariance stabilization)
**Ensure:** Final model outputs $\boldsymbol{Y}$ for all test inputs

1: **Approximating the distributions as Gaussian:**              ▷ One-time execution
2: **for** $t = 1$ to $T$ **do**
3:     Extract features:
4:         $f_{\mathrm{merge}}(\bar{\boldsymbol{x}}_t^{(i)}) = f(\bar{\boldsymbol{x}}_t^{(i)}; \boldsymbol{\theta}_{\mathrm{merge}}), \quad i = 1, \ldots, N.$
5:     Compute $\hat{\boldsymbol{\mu}}_t$ and $\hat{\boldsymbol{\Sigma}}_t$ as in Eq. 2 and Eq. 3
6:     Add $\epsilon$ to diagonal of $\hat{\boldsymbol{\Sigma}}_t$ to stabilize inversion.
7: **end for**

8: **Group batch inference:**              ▷ Test time (Inference)
9: initialize output $\boldsymbol{Y} \leftarrow \{\}$
10: **for** each test batch $\acute{\boldsymbol{X}}$ **do**
11:     **for** $t = 1$ to $T$ **do**
12:         Compute $M_t(\acute{\boldsymbol{X}})$ as in Eq. 4
13:     **end for**
14:     $\hat{\boldsymbol{\tau}} = \arg\min_t M(\acute{\boldsymbol{X}})$
15:     Grouping of $\acute{\boldsymbol{X}}$ by classified tasks $\hat{\boldsymbol{\tau}}$
16:     **for** $\hat{\tau} = 1$ to $|\hat{\boldsymbol{\tau}}|$ **do**
17:         $\boldsymbol{\theta}_{\mathrm{merge}}^{\star} \leftarrow \boldsymbol{\theta}_0 + \mathcal{M}_{\hat{\tau}}(\boldsymbol{\Delta}_{\mathrm{merge}})$
18:         Forward pass for the $\hat{\tau}$-th input group $\boldsymbol{X}_{\hat{\tau}}'$ using $\boldsymbol{\theta}_{\mathrm{merge}}^{\star}$: $f(\acute{\boldsymbol{X}}_{\hat{\tau}}; \boldsymbol{\theta}_{\mathrm{merge}}^{\star})$
19:         $\boldsymbol{Y} \leftarrow \boldsymbol{Y} \cup f(\acute{\boldsymbol{X}}_{\hat{\tau}}; \boldsymbol{\theta}_{\mathrm{merge}}^{\star}).$
20:     **end for**
21: **end for**

---

## C  EXPERIMENTAL DETAILS

### C.1  TRAINING DETAILS

**Computer vision tasks.** All experiments have followed the training methodology outlined by TALL-Mask (Wang et al., 2024). We have employed the identical pre-trained CLIP ViT (Dosovitskiy et al., 2021; Radford et al., 2021) checkpoint sourced from the open_clip repository (Ilharco et al., 2023), fine-tuning for 2,000 iterations. The dataset has been split into training and validation sets, comprising 90% and 10% of the data, respectively. The training configuration included a batch size of 128, a learning rate of $10^{-5}$ (managed by a cosine annealing schedule with 200 warm-up steps), the AdamW optimizer, and cross-entropy as the loss function. Furthermore, in accordance

with Task Arithmetic (Ilharco et al., 2023) and TALL-Mask (Wang et al., 2024), the classification layer weights have remained fixed throughout fine-tuning.

**NLP tasks.** Following TIES-Merging (Yadav et al., 2023), we have employed T5-large (Raffel et al., 2020) as the pre-trained backbone sourced from the `HuggingFace`, which has been trained for up to 75,000 steps with an effective batch size of 1024 and a learning rate of 0.0001. Then, we fine-tuned T5-large (Raffel et al., 2020) independently on seven tasks.

Note that, unlike TWIN-Merging (Lu et al., 2024), our proposed method MAD does not require router training and therefore incurs no training time.

## C.2 BASELINE DETAILS

We now describe the baselines utilized for our main comparison experiment:

- **Fine-tuned**: This baseline represents a fine-tuned task-specific model for each task. These models are inherently single-task (i.e., they cannot execute multiple tasks concurrently).

- **Weight Averaging** (Shoemake, 1985; Utans, 1996): This baseline represents the most fundamental approach to weight interpolation, directly averaging parameters from multiple models under the assumption of uniform importance (i.e., equal weighting) for each task-specific model.

- **Task Arithmetic** (Ilharco et al., 2023): This baseline first computes *task vectors* $\boldsymbol{\tau}_t$ for each task $t$ by determining the difference between its fine-tuned parameters $\boldsymbol{\theta}_t$ and the pre-trained parameters $\boldsymbol{\theta}_0$ (i.e., $\boldsymbol{\tau}_t = \boldsymbol{\theta}_t - \boldsymbol{\theta}_0$). A merged model $\boldsymbol{\theta}_{\mathrm{merge}}$ is then formed by adding a linear combination of these task vectors to the pre-trained parameters. This combination is expressed as $\boldsymbol{\theta}_{\mathrm{merge}} = \boldsymbol{\theta}_0 + \alpha \sum_{t=1}^{T} \boldsymbol{\tau}_t$, where each task vector $\boldsymbol{\tau}_t$ is scaled by a hyper-parameter $\alpha$. The hyper-parameter $\alpha$ is tuned from the set $\{0.0, 0.1, ..., 0.9, 1.0\}$ by optimizing for the average performance across all tasks on their respective validation sets.

- **TIES-Merging** (Yadav et al., 2023): This baseline performs weight interpolation through three stages: *Trim*, *Elect Sign*, and *Merge*. During the Trim stage, all task vector values are set to zero except for the top 20% based on magnitude. The Merge stage then proceeds identically to Task Arithmetic, with hyper-parameter tuning conducted as described previously.

- **TALL Mask** (Wang et al., 2024): This baseline initially employs a task vector of multi-task model $\boldsymbol{\tau}_{\mathrm{MTL}}$ to generate masks that identify parameters crucial for each task. Typically, the multi-task model is created using Task Arithmetic (TA) or TIES-Merging (TIES). When using TA, the scaling factor $\alpha$ is tuned within $\{0.0, 0.1, \ldots, 1.0\}$ and selected according to the average validation performance over all tasks. Subsequently, a *task localization masks* $\boldsymbol{m}_t$ is created, targeting to construct $\hat{\boldsymbol{\theta}}_t$ such that: $\hat{\boldsymbol{\theta}}_t = \boldsymbol{\theta}_0 + \boldsymbol{m}_t \circ \boldsymbol{\tau}_{MTL}$. The mask is obtained by minimizing the $\ell_1$ distance between the reconstructed $\hat{\theta}_t$ and fine-tuned parameters $\theta_t$. The generation of these masks is governed by a hyper-parameter $\lambda_t$ for task $t$, which is tuned over the range $\{0.2, 0.3, 0.4, 0.5, 0.6\}$ based on validation set performance.

- **EMR-Merging** (Huang et al., 2024): This baseline first obtains an aggregated *unified task vector* $\boldsymbol{\tau}_{\mathrm{uni}}$ by leveraging the sign and absolute value of each individual task vector. Subsequently, it computes a task-specific mask $M_t$ and rescaler $\lambda_t$ for each task. The task-specific mask and rescaler are adjusted according to the particular evaluation task. Finally, weight interpolation is achieved by element-wise multiplying the unified task vector with the respective task-specific masks and rescalers: $\boldsymbol{\theta}_{\mathrm{merge}} = \boldsymbol{\theta}_0 + \lambda_t \cdot M_t \odot \boldsymbol{\tau}_{\mathrm{uni}}$. Notably, this baseline does not involve any hyper-parameter tuning.

- **TWIN-Merging** (Lu et al., 2024): This baseline initially trains a router $\mathcal{R}(\cdot; \phi)$, parameterized by $\phi$, to enable effective task classification. Subsequently, a *shared expert* $f(\cdot; \boldsymbol{\theta}_s)$ parameterized by $\boldsymbol{\theta}_s$ is extracted by Task Arithmetic with fixed hyper-parameter ($\alpha = 0.29$). Following this, *exclusive knowledge vectors* $\boldsymbol{v}_t$ are extracted for each task, employing either Singular Value Decomposition (SVD) or the Trim procedure from TIES-Merging. In this comparison, Trim demonstrates the best performance. During test time,

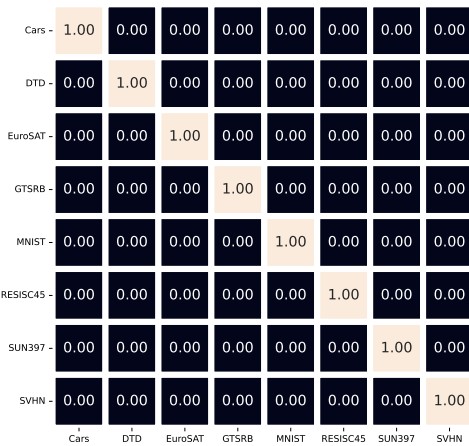

Figure D1: **Pair-wise Bhattacharyya distance (left) and Bhattacharyya coefficient (right) between task-conditional Gaussian feature distributions of different tasks.** The heatmaps illustrate these metrics calculated from the features extracted by the merged model.

the router determines task-specific weights $w_t = \mathrm{softmax}(\mathcal{R}(\mathrm{Emb}(\boldsymbol{x}); \phi))$ for each input $\boldsymbol{x}$, where $\mathrm{Emb}(\boldsymbol{x})$ denotes the embeddings of the penultimate layer from the shared expert. Weight interpolation is then performed by adding to the shared expert a weighted sum of these exclusive knowledge vectors, where the weights are the routed task-specific weights: $\boldsymbol{\theta}_{\mathrm{merge}} = \boldsymbol{\theta}_s + \sum_{t=1}^{T} w_t \boldsymbol{v}_t$.

- **DaWin** (Oh et al., 2025): This baseline performs weight interpolation by assigning a weight to each task for every input. This weight $\lambda_t(\boldsymbol{x})$ for task $t$ is derived from an entropy calculation that considers the task-specific model for that input $x$. Weight interpolation is then performed according to the following equation: $\boldsymbol{\theta}_{\mathrm{merge}} = \boldsymbol{\theta}_0 + \alpha \sum_{t=1}^{T} \lambda_t(\boldsymbol{x}) \boldsymbol{\tau}_t$, where $\alpha$ is fixed by $0.3$. Additionally, to reduce runtime overhead during inference, a Beta mixture model (BMM) can be additionally employed. The default number of mixture components ($K$) for BMM is 3.

## C.3 INFERENCE DETAILS

For all inference procedures, including performance evaluation and computational cost measurement, experiments involving 14 tasks or fewer have been conducted using NVIDIA GeForce RTX 3090 GPUs, while those with 20 or more tasks have utilized NVIDIA H100 80GB HBM3 GPUs.

# D FURTHER DISCUSSION ON MAHALANOBIS-DISTANCE-BASED TASK CLASSIFICATION

Continuing from Section 4.1, this section further discusses the suitability of Mahalanobis distance for task classification. In MAD, the mean and covariance of features are computed from training set samples, effectively modeling these features with a Gaussian distribution. These statistics are subsequently utilized during test time to compute Mahalanobis distances for task classification. Consequently, the justification for using Mahalanobis distance in task classification under Gaussian discriminant analysis (GDA) hinges on the separability of the approximated Gaussian feature distributions for each task.

To evaluate the degree of separation between the approximated Gaussian feature distributions across tasks, we utilize the Bayes error bound. As our task classifier is defined under GDA, the task classifier within MAD can be considered a Bayes classifier. A Bayes classifier is defined as follows:

**Definition 2** (Bayes classifier). *Let $\mathcal{X}$ be the input space and $\mathcal{T} = \{1, 2, \dots, T\}$ the set of task indices. Given the joint data-generating distribution $p(\boldsymbol{x}, \tau = t) = p(\tau = t)p(\boldsymbol{x}|\tau = t)$, the **Bayes***

***classifier*** $\mathcal{R}^\star : \mathcal{X} \to \mathcal{T}$ *is the decision rule that minimizes the expected 0-1 loss, namely*

$$\hat{\tau} = \mathcal{R}^\star(\boldsymbol{x}) = \arg\max_{t \in \mathcal{T}} \; p(\tau = t) p(\boldsymbol{x} | \tau = t),$$

*where $p(\boldsymbol{x}|\tau = t)$ is the task-conditional likelihood and $p(\tau = t)$ is the prior probability of task $t$.*

The error of this Bayes classifier and its bound are defined as follows:

**Definition 3** (Bayes error bound). *The **Bayes error** $\mathcal{E}_{Bayes}$ is the minimum achievable misclassification probability achieved by the Bayes classifier $\mathcal{R}^\star$. This error is defined by the probability mass contained in the overlap of the class-conditional densities; formally, for $\mathcal{T} = \{1, \ldots, T\}$:*

$$\mathcal{E}_{Bayes} = \int_{\mathcal{X}} \min_{t \in \mathcal{T}} p(\tau = t) p(\boldsymbol{x} | \tau = t) \, dx,$$

*where the integrand chooses, at each $\boldsymbol{x}$, all but the largest class-weighted density, thus integrating precisely the region where different tasks interfere.*

*Since the true densities $p(\boldsymbol{x}|\tau = t)$ are seldom known exactly, $\mathcal{E}_{Bayes}$ is generally intractable in closed form. Consequently, one resorts to **an upper bound** $\mathcal{B}$ that is analytically computable from pair-wise divergences or distance measures, guaranteeing*

$$\mathcal{E}_{Bayes} \le \mathcal{B}.$$

According to Definition 3, the Bayes error quantifies the degree of overlap between the feature distributions of different tasks. Thus, a lower Bayes error implies that a clearer decision boundary can be established between the features of these tasks, indicating that they are clearly separable. This, in turn, suggests that the features exhibit a distribution more amenable to classification.

Given that the Bayes error is generally intractable, an upper bound must be established for it. Therefore, we utilize the Bhattacharyya distance as a surrogate for this bound. Formally, we first define the Bhattacharyya distance and then state the following theorem:

**Definition 4** (Bhattacharyya distance). *Let $\mathcal{X}$ be the input space and let $P$ and $Q$ be two probability measures on $\mathcal{X}$ with densities $p(\boldsymbol{x})$ and $q(\boldsymbol{x})$ with respect to a common base measure.*

*The **Bhattacharyya distance** $B(P, Q)$ is the divergence*

$$B(P, Q) = -\ln \int_{\mathcal{X}} \sqrt{p(\boldsymbol{x})q(\boldsymbol{x})} \, dx.$$

*For class-conditional densities $p(\boldsymbol{x}|\tau = t)$ and $p(\boldsymbol{x}|\tau = s)$ with task labels $t, s \in \mathcal{T}$, the distance is is given by*

$$B(t, s) = -\ln \int_{\mathcal{X}} \sqrt{p(\boldsymbol{x}|\tau = t)p(\boldsymbol{x}|\tau = s)} \, dx.$$

*Assume the class-conditional feature densities are multivariate Gaussians*

$$p(\boldsymbol{x}|\tau = t) = \mathcal{N}(\boldsymbol{x}|\boldsymbol{\mu}_t, \boldsymbol{\Sigma}_t), \quad p(\boldsymbol{x}|\tau = s) = \mathcal{N}(\boldsymbol{x}|\boldsymbol{\mu}_s, \boldsymbol{\Sigma}_s).$$

*Then the Bhattacharyya distance admits the closed form*

$$B(t, s) = \frac{1}{8}(\boldsymbol{\mu}_t - \boldsymbol{\mu}_s)^\top \boldsymbol{\Sigma}_{ts}^{-1}(\boldsymbol{\mu}_t - \boldsymbol{\mu}_s) + \frac{1}{2}\ln\frac{\det \boldsymbol{\Sigma}_{ts}}{\sqrt{\det \boldsymbol{\Sigma}_t \det \boldsymbol{\Sigma}_s}}, \quad \boldsymbol{\Sigma}_{ts} = \frac{1}{2}(\boldsymbol{\Sigma}_t + \boldsymbol{\Sigma}_s). \quad (7)$$

**Theorem 1** (Bayes error is bounded by Bhattacharyya distance (Bhattacharyya, 1943; Kailath, 1967; Fukunaga, 1990)). *Let $t, s \in \mathcal{T}$ be two classes with priors $p(\tau = t), p(\tau = s) > 0$ $(p(\tau = t) + p(\tau = s) = 1)$. Assume each class-conditional density is multivariate Gaussian. Then the Bhattacharyya distance $B(t, s)$ defined in Equation 7 yields the Bayes error bound*

$$\mathcal{E}_{Bayes} \le \sqrt{p(\tau = t)p(\tau = s)} \exp[-B(t, s)]. \quad (8)$$

The proof for Theorem 1 is detailed in Section E.

Figure D1 visualizes the separability and similarity of Gaussian feature distributions for eight computer vision tasks, as described in Section 5.1. The Bhattacharyya distance generally increases for off-diagonal elements from the first layer to the last layer; moreover, the Bhattacharyya coefficient, as defined in Equation 11, for these same elements significantly decreases. This suggests that when the feature distributions from the merged model are assumed to be Gaussian, they are indeed well-separated. Therefore, these findings validate the approach of assuming Gaussian feature distributions and performing task classification using Mahalanobis distance within the GDA framework. Furthermore, this analysis may also offer insight into why MAD demonstrates superior performance than other model merging methods.

## E  PROOF FOR THEOREM 1

We provide a proof for Theorem 1 in this section.

*Proof.* The Bayes error for two classes is

$$\mathcal{E}_{\text{Bayes}} = \int_{\mathcal{X}} \min[p(\tau = t)p(\boldsymbol{x}|\tau = t), \ p(\tau = s)p(\boldsymbol{x}|\tau = s)] \, dx. \tag{9}$$

For any non-negative $a, b$ the elementary inequality $\min(a, b) \le \sqrt{ab}$ holds.

Applying this to the integrand in (9) gives

$$\mathcal{E}_{\text{Bayes}} \le \sqrt{p(\tau = t)p(\tau = s)} \int_{\mathcal{X}} \sqrt{p(\boldsymbol{x}|\tau = t)p(\boldsymbol{x}|\tau = s)} \, dx = \sqrt{p(\tau = t)p(\tau = s)} \, \text{BC}(t, s), \tag{10}$$

where

$$\text{BC}(t, s) := \int_{\mathcal{X}} \sqrt{p(\boldsymbol{x}|\tau = t)p(\boldsymbol{x}|\tau = s)} \, dx \tag{11}$$

is the *Bhattacharyya coefficient*.

Write each density

$$p(\boldsymbol{x}|\tau) = (2\pi)^{\frac{d}{2}} (\det \boldsymbol{\Sigma}_\tau)^{-\frac{1}{2}} \exp\left[-\frac{1}{2}(\boldsymbol{x} - \boldsymbol{\mu}_\tau)^\top \boldsymbol{\Sigma}_\tau^{-1}(\boldsymbol{x} - \boldsymbol{\mu}_\tau)\right].$$

Taking square-roots and multiplying:

$$\sqrt{p(\boldsymbol{x}|\tau = t)p(\boldsymbol{x}|\tau = s)}$$

$$= (2\pi)^{-\frac{d}{2}} (\det \boldsymbol{\Sigma}_t \det \boldsymbol{\Sigma}_s)^{-\frac{1}{4}} \exp\left[-\frac{1}{4}\left((\boldsymbol{x} - \boldsymbol{\mu}_t)^\top \boldsymbol{\Sigma}_t^{-1}(\boldsymbol{x} - \boldsymbol{\mu}_t) + (\boldsymbol{x} - \boldsymbol{\mu}_s)^\top \boldsymbol{\Sigma}_s^{-1}(\boldsymbol{x} - \boldsymbol{\mu}_s)\right)\right]. \tag{12}$$

Rewrite the exponent in (12):

$$(\boldsymbol{x} - \boldsymbol{\mu}_t)^\top \boldsymbol{\Sigma}_t^{-1}(\boldsymbol{x} - \boldsymbol{\mu}_t) + (\boldsymbol{x} - \boldsymbol{\mu}_s)^\top \boldsymbol{\Sigma}_s^{-1}(\boldsymbol{x} - \boldsymbol{\mu}_s)$$
$$= 2(\boldsymbol{x} - m)^\top \boldsymbol{\Sigma}_{ts}^{-1}(\boldsymbol{x} - m) + (\boldsymbol{\mu}_t - \boldsymbol{\mu}_s)^\top \boldsymbol{\Sigma}_{ts}^{-1}(\boldsymbol{\mu}_t - \boldsymbol{\mu}_s), \tag{13}$$

where the *Bhattacharyya mean*

$$m := \boldsymbol{\Sigma}_{ts}\left(\frac{1}{2}\boldsymbol{\Sigma}_t^{-1}\boldsymbol{\mu}_t + \frac{1}{2}\boldsymbol{\Sigma}_s^{-1}\boldsymbol{\mu}_s\right).$$

Substituting (13) into (12) and integrating:

$$\text{BC}(t, s) = (2\pi)^{-\frac{d}{2}} (\det \boldsymbol{\Sigma}_t \det \boldsymbol{\Sigma}_s)^{-\frac{1}{4}} \exp\left[-\frac{1}{8}(\boldsymbol{\mu}_t - \boldsymbol{\mu}_s)^\top \boldsymbol{\Sigma}_{ts}^{-1}(\boldsymbol{\mu}_t - \boldsymbol{\mu}_s)\right]$$

$$\times \int_{\mathcal{X}} \exp\left[-\frac{1}{2}(\boldsymbol{x} - m)^\top \boldsymbol{\Sigma}_{ts}^{-1}(\boldsymbol{x} - m)\right] \, dx$$

$$= (2\pi)^{-\frac{d}{2}} (\det \boldsymbol{\Sigma}_t \det \boldsymbol{\Sigma}_s)^{-\frac{1}{4}} \exp\left[-\frac{1}{8}(\boldsymbol{\mu}_t - \boldsymbol{\mu}_s)^\top \boldsymbol{\Sigma}_{ts}^{-1}(\boldsymbol{\mu}_t - \boldsymbol{\mu}_s)\right]$$

$$\times (2\pi)^{\frac{d}{2}} (\det \boldsymbol{\Sigma}_{ts})^{\frac{1}{2}}. \tag{14}$$

Cancelling the $(2\pi)^{\pm\frac{d}{2}}$ prefactors yields

$$\mathrm{BC}(t,s) = \frac{\det(\boldsymbol{\Sigma}_{ts})^{\frac{1}{2}}}{(\det \boldsymbol{\Sigma}_t \det \boldsymbol{\Sigma}_s)^{\frac{1}{4}}} \exp\left[-\frac{1}{8}(\boldsymbol{\mu}_t - \boldsymbol{\mu}_s)^\top \boldsymbol{\Sigma}_{ts}^{-1}(\boldsymbol{\mu}_t - \boldsymbol{\mu}_s)\right]. \quad (15)$$

Taking minus the logarithm of (15) gives exactly the distance $B(t,s)$ as defined in the Definition 4:

$$\mathrm{BC}(t,s) = \exp[-B(t,s)]. \quad (16)$$

Substituting (16) into (10):

$$\mathcal{E}_{\mathrm{Bayes}} \le \sqrt{p(\tau = t)p(\tau = s)} \exp[-B(t,s)],$$

which is Inequality (8). □

## F ADDITIONAL RESULTS

**Varying seeds and sample sizes.** Table F1 presents an ablation study investigating the sensitivity of MAD to two key factors: the number of samples $N$ and the random seed used for initialization. According to these results, MAD demonstrates robustness against randomness, such as changes in seed initialization, and performs well even when utilizing a very small number of samples per task (64 or fewer). While performance improves with increasing $N$, the magnitude of improvement appears to diminish, especially after $N = 16$ or $N = 32$. The jump from $N = 4$ to $N = 8$ shows a notable increase in average accuracy. However, the gains from $N = 16$ to $N = 32$, and further to $N = 64$, are more marginal, suggesting that using a moderate number of samples ($N \ge 16$) might be sufficient to capture most of the performance benefits. On the other hand, as $N$ increases, the impact of the random seed diminishes significantly. For $N = 16, 32, 64$, the average accuracies across different seeds are very stable. For individual tasks as well, the variance due to seed choice becomes negligible at higher $N$. The choice of $N = 64$ (as potentially indicated by "default") appears to be a robust setting, providing consistently high performance with minimal variance due to seed initialization. These results imply that MAD can be particularly valuable in scenarios where data acquisition is challenging or where robustness to random variations is essential.

**Varying seeds and epsilon values.** Table F2 presents an ablation study investigating the sensitivity of MAD to two key factors: the value of epsilon $\epsilon$ and the random seed used for initialization. According to these results, MAD demonstrates strong robustness across different epsilon values and consistently stable performance irrespective of the random seed used. Specifically, our analysis shows that even with diverse $\epsilon$ settings, the average accuracy remains high, with differences typically being marginal. Furthermore, the impact of the random seed on performance is consistently negligible. For all tested $\epsilon$ values, the average accuracies across different seeds are remarkably stable. This stability extends to individual tasks as well, where the variance due to seed choice becomes negligible. These findings collectively underscore the practical reliability of MAD, confirming its consistent high performance and minimal sensitivity to common sources of variability.

**Task classification performance.** To further examine whether the Gaussian assumption in our task-identification module is reasonable in practice, we compare the task–classification accuracy of different routing strategies in Table F3. The table reports the average task–ID prediction accuracy across three CLIP ViT backbones (ViT-B/32, ViT-B/16, and ViT-L/14) and three task configurations (8, 14, and 20 tasks), comparing our MAD (MAD, which fits a Gaussian distribution and computes Mahalanobis distance in the feature space) against a cosine-distance $k$-NN baseline (no explicit distributional assumption) and the learned MLP-based router of TWIN-Merging.

For a controlled comparison in the *training-free* setting, both MAD and $k$-NN use only 64 samples per task as calibration data, whereas TWIN-Merging trains its router on a substantially larger labeled set of 1000 samples per task. Despite this advantage in sample size and model capacity, MAD consistently achieves the highest task–classification accuracy across all backbones and task counts. In particular, MAD outperforms $k$-NN in every configuration, indicating that modeling features with a Gaussian distribution and using Mahalanobis distance is more effective than a purely non-parametric

Table F1: **Ablation study on the impact of the sample size ($N$) and random seed initialization on the performance of MAD using a CLIP ViT-B/32.** Accuracy is reported across eight vision datasets. The table shows results for $N = \{8, 16, 32, 64\}$, with five different random seeds (0-4) for each $N$. Seed 0 is considered the default.

| Method (ViT-B/32) | Cars | DTD | EuroSAT | GTSRB | MNIST | RESISC45 | SUN397 | SVHN | Avg. |
|---|---|---|---|---|---|---|---|---|---|
| | | | | $N = 4$ | | | | | |
| MAD (seed: 0) | 99.8 | 93.8 | 87.5 | 98.3 | 95.2 | 82.1 | 82.5 | 84.4 | 90.4 |
| MAD (seed: 1) | 99.9 | 91.3 | 75.4 | 94.5 | 90.9 | 63.5 | 89.5 | 94.6 | 87.4 |
| MAD (seed: 2) | 99.7 | 90.9 | 88.1 | 92.4 | 97.1 | 69.8 | 71.0 | 88.4 | 87.2 |
| MAD (seed: 3) | 97.3 | 97.0 | 68.9 | 88.8 | 88.1 | 80.5 | 81.8 | 97.5 | 87.5 |
| MAD (seed: 4) | 99.9 | 94.4 | 70.9 | 96.8 | 93.8 | 89.3 | 83.4 | 93.3 | 90.2 |
| | | | | $N = 8$ | | | | | |
| MAD (seed: 0) | 99.8 | 96.8 | 93.3 | 95.5 | 98.3 | 88.9 | 83.8 | 91.8 | 93.5 |
| MAD (seed: 1) | 99.8 | 92.5 | 84.0 | 97.2 | 96.4 | 79.6 | 91.8 | 94.0 | 91.9 |
| MAD (seed: 2) | 99.7 | 91.7 | 98.7 | 99.0 | 91.1 | 86.1 | 93.8 | 91.5 | 94.0 |
| MAD (seed: 3) | 99.7 | 96.9 | 81.7 | 96.8 | 97.9 | 85.2 | 82.1 | 90.6 | 91.4 |
| MAD (seed: 4) | 99.9 | 94.2 | 89.9 | 95.8 | 99.4 | 85.4 | 91.3 | 93.1 | 93.6 |
| | | | | $N = 16$ | | | | | |
| MAD (seed: 0) | 99.9 | 94.4 | 99.6 | 96.7 | 99.2 | 92.4 | 88.6 | 96.0 | 95.9 |
| MAD (seed: 1) | 99.9 | 93.5 | 97.9 | 98.0 | 87.5 | 87.4 | 91.0 | 96.0 | 93.9 |
| MAD (seed: 2) | 99.8 | 96.0 | 99.0 | 99.5 | 97.6 | 88.0 | 88.7 | 95.6 | 95.5 |
| MAD (seed: 3) | 99.9 | 95.6 | 86.2 | 98.4 | 96.7 | 91.6 | 88.6 | 98.8 | 94.5 |
| MAD (seed: 4) | 99.9 | 97.9 | 93.8 | 96.9 | 99.1 | 89.8 | 87.5 | 96.0 | 95.1 |
| | | | | $N = 32$ | | | | | |
| MAD (seed: 0) | 99.9 | 97.6 | 99.8 | 98.3 | 99.1 | 94.8 | 88.7 | 96.1 | 96.8 |
| MAD (seed: 1) | 99.9 | 93.5 | 99.8 | 99.0 | 93.9 | 94.0 | 90.8 | 96.6 | 95.9 |
| MAD (seed: 2) | 99.9 | 96.0 | 97.4 | 99.1 | 98.7 | 92.9 | 89.8 | 97.6 | 96.4 |
| MAD (seed: 3) | 99.9 | 95.7 | 95.1 | 98.9 | 99.6 | 96.9 | 87.9 | 99.0 | 96.6 |
| MAD (seed: 4) | 99.9 | 98.1 | 97.8 | 97.3 | 98.7 | 91.1 | 89.1 | 95.1 | 95.9 |
| | | | | $N = 64$ | | | | | |
| **MAD (seed: 0, default)** | 99.9 | 98.4 | 99.9 | 98.8 | 99.9 | 96.6 | 92.1 | 98.6 | 98.0 |
| MAD (seed: 1) | 99.9 | 97.6 | 99.8 | 99.1 | 99.9 | 95.6 | 93.2 | 98.6 | 98.0 |
| MAD (seed: 2) | 99.9 | 97.5 | 99.6 | 99.5 | 100 | 97.4 | 94.3 | 99.3 | 98.4 |
| MAD (seed: 3) | 99.9 | 97.4 | 98.8 | 98.5 | 100 | 97.5 | 91.1 | 99.5 | 97.8 |
| MAD (seed: 4) | 99.9 | 98.0 | 99.3 | 99.2 | 99.9 | 97.3 | 90.2 | 97.9 | 97.7 |

nearest-neighbor approach. Moreover, MAD yields sizable margins over TWIN-Merging (often exceeding 10 percentage points), even though TWIN-Merging relies on a learned router with many more labeled examples. These observations suggest that, even when the true feature distribution may be complex or unknown, the Gaussian approximation adopted by MAD already provides a strong and practical mechanism for task identification without additional training, while exploring more sophisticated non-Gaussian models remains an interesting direction for future work.

**Scalability to Larger Task Sets.** To assess scalability to larger and more heterogeneous task collections, we additionally evaluate task classification on 30- and 50-task vision suites using a CLIP ViT-B/32 backbone. As shown in Figure F1, even with an increased number of tasks (up to 50), although some overlaps observe in domains sharing inherent similarities, the majority of task clusters remain well-separated. Furthermore, As summarized in Table F4, MAD maintains strong task–ID prediction accuracy and consistently outperforms all $k$-NN variants as well as the TWIN-Merging router by substantial margins. Notably, even when scaling to 50 tasks, MAD exceeds the TWIN router by more than 30 percentage points. Crucially, in Table F5 MAD also demonstrates the best overall multi-task performance for both 30- and 50-task scenarios. Taken together with the 8–20 task results, these findings indicate that the Gaussian approximation and Mahalanobis-distance-based routing used in MAD are robust and scalable as the number of tasks grows, while still requiring no additional training beyond a small calibration set.

**Performance on Fine-Grained and Similar Tasks.** To rigorously assess robustness of MAD when faced with highly similar and fine-grained tasks, we have conducted an experiment utilizing the CUB-200 birds dataset. To ensure distinct task definitions while maintaining fine-grained distinctions, we split the CUB-200 dataset into 8 separate classes, each serving as an individual

Table F2: **Ablation study on the impact of the epsilon values ($\epsilon$) and random seed initialization on the performance of MAD using a CLIP ViT-B/32.** Accuracy is reported across eight vision datasets. The table shows results for $\epsilon = \{1e^{-2}, 1e^{-3}, 1e^{-4}, 1e^{-5}, 1e^{-6}\}$, with five different random seeds (0-4) for each $N$. Seed 0 is considered the default.

| Method (ViT-B/32) | Cars | DTD | EuroSAT | GTSRB | MNIST | RESISC45 | SUN397 | SVHN | Avg. |
|---|---|---|---|---|---|---|---|---|---|
| | | | | $\epsilon = 1e^{-2}$ | | | | | |
| MAD (seed: 0) | 99.9 | 98.5 | 99.8 | 99.2 | 100 | 96.6 | 96.8 | 97.1 | 98.5 |
| MAD (seed: 1) | 99.8 | 97.7 | 99.2 | 99.2 | 99.9 | 96.3 | 96.5 | 97.0 | 98.2 |
| MAD (seed: 2) | 99.9 | 97.3 | 98.6 | 99.6 | 100 | 97.4 | 96.9 | 97.8 | 98.4 |
| MAD (seed: 3) | 99.9 | 97.5 | 97.8 | 99.2 | 100 | 96.8 | 96.0 | 98.7 | 98.2 |
| MAD (seed: 4) | 99.9 | 98.2 | 97.9 | 99.5 | 99.9 | 97.7 | 94.8 | 96.6 | 98.1 |
| | | | | $\epsilon = 1e^{-3}$ | | | | | |
| MAD (seed: 0) | 99.9 | 98.6 | 99.9 | 99.0 | 99.9 | 96.7 | 93.9 | 98.3 | 98.3 |
| MAD (seed: 1) | 99.9 | 97.7 | 99.6 | 99.2 | 99.9 | 96.1 | 94.4 | 98.3 | 98.1 |
| MAD (seed: 2) | 99.9 | 97.5 | 99.1 | 99.5 | 100 | 97.6 | 95.4 | 99.0 | 98.5 |
| MAD (seed: 3) | 99.9 | 97.4 | 98.4 | 98.9 | 100 | 97.6 | 93.1 | 99.4 | 98.1 |
| MAD (seed: 4) | 99.9 | 98.4 | 98.9 | 99.5 | 99.9 | 97.6 | 91.9 | 97.7 | 98.0 |
| | | | | $\epsilon = 1e^{-4}$ | | | | | |
| MAD (seed: 0) | 99.9 | 98.4 | 98.9 | 98.8 | 99.9 | 96.6 | 92.1 | 98.6 | 98.0 |
| MAD (seed: 1) | 99.9 | 97.6 | 99.8 | 99.2 | 99.9 | 95.7 | 93.2 | 98.6 | 98.0 |
| MAD (seed: 2) | 99.9 | 97.6 | 99.7 | 99.5 | 100 | 97.4 | 94.3 | 99.3 | 98.5 |
| MAD (seed: 3) | 99.9 | 97.4 | 98.9 | 98.6 | 100 | 97.6 | 91.1 | 99.5 | 97.7 |
| MAD (seed: 4) | 99.9 | 98.0 | 99.4 | 99.3 | 99.9 | 97.3 | 90.3 | 98.0 | 97.8 |
| | | | | $\epsilon = 1e^{-5}$ | | | | | |
| MAD (seed: 0) | 99.9 | 98.4 | 99.9 | 98.7 | 99.9 | 96.5 | 91.7 | 98.6 | 98.0 |
| MAD (seed: 1) | 99.9 | 97.6 | 99.8 | 99.1 | 99.9 | 95.4 | 93.0 | 98.6 | 98.0 |
| MAD (seed: 2) | 99.9 | 97.6 | 99.7 | 99.5 | 100 | 97.4 | 94.1 | 99.3 | 98.4 |
| MAD (seed: 3) | 99.9 | 97.2 | 98.9 | 98.5 | 100 | 97.4 | 90.7 | 99.5 | 97.8 |
| MAD (seed: 4) | 99.9 | 98.0 | 99.4 | 99.1 | 99.9 | 97.2 | 89.9 | 97.9 | 97.7 |
| | | | | $\epsilon = 1e^{-6}$ | | | | | |
| **MAD (seed: 0, default)** | 99.9 | 98.0 | 99.4 | 99.1 | 99.9 | 97.2 | 89.9 | 97.9 | 97.7 |
| MAD (seed: 1) | 99.9 | 97.6 | 99.8 | 99.1 | 99.9 | 95.3 | 92.9 | 98.6 | 97.9 |
| MAD (seed: 2) | 99.9 | 97.5 | 99.7 | 99.5 | 100 | 97.4 | 94.1 | 99.3 | 98.5 |
| MAD (seed: 3) | 99.9 | 97.2 | 98.9 | 98.4 | 100 | 97.4 | 90.7 | 99.5 | 97.7 |
| MAD (seed: 4) | 99.9 | 97.9 | 99.4 | 99.1 | 99.9 | 97.2 | 89.9 | 97.9 | 97.7 |

Table F3: **Task classification accuracy (%) of different routing strategies across ViT backbones and numbers of tasks.** We compare our MAD with cosine-distance k-NN and the learned router of TWIN-Merging. MAD and k-NN are both training-free and use only 64 samples per task as calibration data, whereas TWIN-Merging trains an MLP-based router with 1000 samples per task.

| Backbone | MAD | k-NN (k=1) | k-NN (k=3) | k-NN (k=5) | k-NN (k=10) | TWIN-Merging |
|---|---|---|---|---|---|---|
| | | | 8 tasks | | | |
| ViT-B/32 | **98.0** | 93.7 | 92.5 | 91.7 | 89.4 | 90.5 |
| ViT-B/16 | **98.3** | 94.7 | 93.6 | 92.8 | 91.3 | 86.8 |
| ViT-L/14 | **98.5** | 93.2 | 92.8 | 92.2 | 90.7 | 83.7 |
| | | | 14 tasks | | | |
| ViT-B/32 | **96.9** | 92.9 | 92.0 | 91.2 | 89.1 | 84.0 |
| ViT-B/16 | **97.0** | 93.4 | 92.4 | 91.7 | 90.2 | 79.9 |
| ViT-L/14 | **97.7** | 93.7 | 93.5 | 93.2 | 92.0 | 72.6 |
| | | | 20 tasks | | | |
| ViT-B/32 | **94.1** | 90.0 | 89.5 | 89.0 | 86.8 | 75.8 |
| ViT-B/16 | **94.7** | 90.4 | 89.4 | 88.8 | 87.0 | 77.6 |
| ViT-L/14 | **95.8** | 91.3 | 91.2 | 91.3 | 90.2 | 78.8 |

task. Parameters for each of these task-specific datasets are then fine-tuned. Our evaluation includes an examination of the feature distributions of samples used for mean and covariance extraction, to assess cluster formation, alongside measurements of both task classification and multi-task performance against existing methods. As expected, shown in Figure F2 the extracted feature distributions for these fine-grained tasks do not form clusters as distinctly separated as those observed with highly

Table F4: **Task classification accuracy (%) on larger 30- and 50-task vision settings using ViT-B/32.** All methods use the same calibration protocol as in Table F3: MAD and $k$-NN are training-free and use 64 samples per task, whereas TWIN-Merging trains an MLP-based router with 1000 samples per task.

| | MAD | k-NN (k=1) | k-NN (k=3) | k-NN (k=5) | k-NN (k=10) | TWIN-Merging |
|---|---|---|---|---|---|---|
| **30 tasks** | **93.6** | 89.2 | 88.9 | 88.6 | 87.1 | 63.1 |
| **50 tasks** | **85.9** | 80.9 | 80.4 | 80.3 | 79.1 | 45.3 |

Table F5: **Multi-task performance of input-dependent merged CLIP ViT models for 30- and 50-computer vision tasks.** We report experimental results for ViT-B/32 on 30 and 50 vision tasks. Bold values represent the best performance among all methods, excluding the individual task-specific baselines.

| Method | 30 tasks | 50 tasks |
|---|---|---|
| Fine-tuned | 93.1 | 90.8 |
| Weight Averaging | 59.1 | 43.8 |
| Task Arithmetic (Task-unknown) | 17.5 | 10.8 |
| TIES-Merging (Task-unknown) | 56.1 | 42.7 |
| TWIN-Merging | 60.1 | 16.4 |
| Dawin | 40.3 | 33.1 |
| WEMoE | 67.1 | 51.6 |
| MoW-Merging | 56.4 | 30.1 |
| **EMR + MAD (Ours)** | 88.1 | 78.6 |
| **TM-TA + MAD (Ours)** | 88.5 | 80.5 |
| **TM-TIES + MAD (Ours)** | **89.6** | **82.5** |

disparate tasks. Nevertheless, as shown in Table F6 MAD still demonstrates superior task classification performance compared to trained Router of TWIN-Merging, despite training-free nature of MAD. This suggests that even when feature distributions exhibit greater overlap, approach of approximating the unknown true distribution with a Gaussian can, in many instances, prove more effective than fitting features to a more complex MLP-based router like TWIN-Merging. Furthermore, when MAD is integrated with existing subspace-based model merging methods, it consistently achieves better multi-task performance than merging methods that do not inherently require a task-unknown scenario assumption. These findings collectively indicate that MAD can be seamlessly applied as an add-on to subspace-based model merging across all task-unknown scenarios (irrespective of whether tasks are distinct or highly similar), providing both improved task classification and enhanced multi-task performance without the need for additional training.

**Effect of feature extraction method on task classification.** To directly investigate whether a good merging method, in terms of overall multi-task performance, necessarily produces features well-suited for task separation, we have conducted an experiment comparing task classification performance using features extracted from models pre-merged with different strategies. Specifically, we have utilized the ViT-B-32 model and evaluated 8 tasks, extracting 64 features per task from models pre-merged using Task Arithmetic, TIES-Merging, and Weight Averaging method. As shown in Figure F3 and F4, our analysis of extracted feature distributions reveals that the Weight Averaging consistently forms the most distinctly separated task clusters, whereas other approaches exhibit noticeable overlaps. This observed overlap in other merging methods such as Task Arithmetic and TIES-Merging, we hypothesize, complicates global task identification and consequently contributes to their diminished task classification performance. Our empirical results in Table F7 further cor-

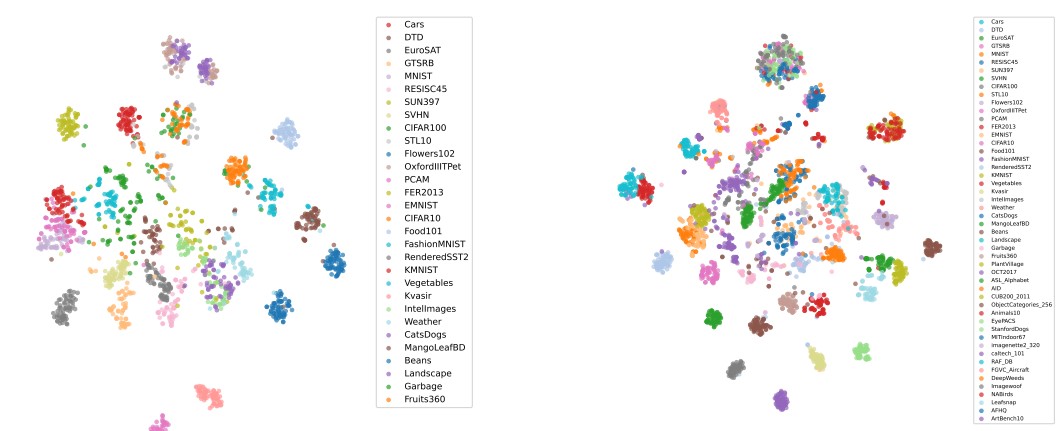

Figure F1: **Task-specific feature distributions from the merged model, approximated by Gaussian, for 30 tasks (left) and 50 tasks (right).**

Table F6: **Performance on 8 fine-grained bird classification tasks from the CUB-200 dataset using ViT-B/32.** This table presents the Multi-task Average Performance (%) for all evaluated methods and the Task Classification Average (%) for methods capable of task identification (MAD and TWIN-Merging). In the table, MAD refers to the performance of TM-TIES + MAD, which showed the best results among EMR + MAD, TM-TA + MAD, and TM-TIES + MAD. A '-' denotes methods that do not perform explicit task classification. Bold values represent the best performance among all methods, excluding the individual task-specific baselines.

| Method | Multi-task Avg. | Task classification Avg. |
|---|---|---|
| Fine-tuned | 78.1 | - |
| Weight Averaging | 60.9 | - |
| Task Arithmetic (Task-unknown) | 51.1 | - |
| TIES-Merging (Task-unknown) | 61.4 | - |
| DaWin | 61.0 | - |
| TWIN-Merging | 64.6 | 55.9 |
| **MAD (Ours)** | **66.9** | **59.4** |

Table F7: **Average Task Classification Accuracy (%) for 8 Tasks Across Different Feature Extraction Methods using ViT-B/32.** This table presents the Mahalanobis distance-based task classification performance using features extracted by various merging methods. For each task, 64 feature samples were utilized for mean and covariance estimation, and the evaluation was performed on 8 tasks.

| Method | Task classification Avg. |
|---|---|
| Weight Averaging | 98.0 |
| Task Arithmetic (Task-unknown) | 92.7 |
| TIES-Merging (Task-unknown) | 90.9 |

roborate this: the Weight Averaging approach achieved the highest task classification performance, thereby directly justifying its selection for feature extraction within MAD.

**Performance on a large language model.** To validate the effectiveness and generalization of our method on a large language model, we have additionally utilized the Llama3.1-8B model. For this evaluation, the training tasks have included (Lambert et al., 2024): Tulu 3 Persona MATH,

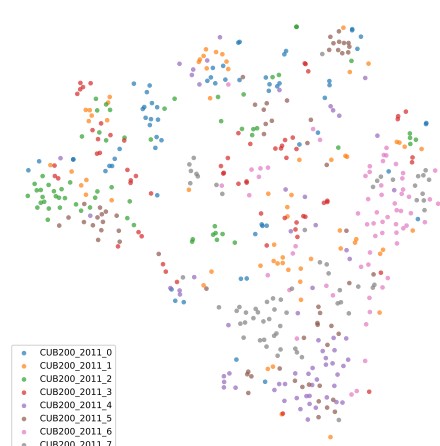

Figure F2: **Task-specific feature distributions from the merged model, approximated as Gaussian distributions, for 8 fine-grained bird classification tasks using ViT-B/32. (CUB-200 dataset).**

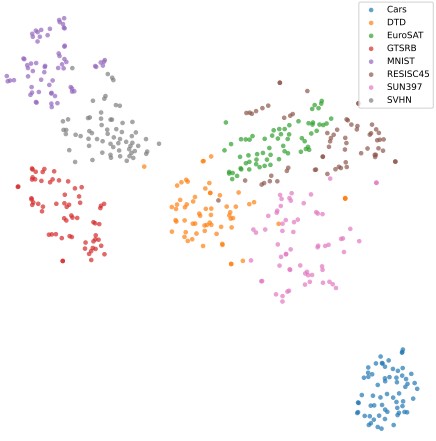

Figure F3: **Task-specific feature distributions from the merged model extracted via Weight Averaging, approximated as Gaussian distributions for 8 tasks.**

OpenMathInstruct 2, and MuminaMath-TIR for the "mathematics reasoning" domain; WildChat, OpenAssistant, and No Robots for the "general instruction following" domain; FLAN v2, SciRIFF, and TableGPT for the "knowledge recall" domain; and Tulu 3 Persona IF for the "precise instruction following" domain. On the other hand, the test tasks have included GSM8K (Cobbe et al., 2021) and MATH (Hendrycks et al., 2021) for the "mathematics reasoning" domain; BBH (CoT) (Suzgun et al., 2023) and DROP (Dua et al., 2019) for the "general instruction following" domain; PopQA (Mallen et al., 2023) for the "knowledge recall" domain; and IFEval (Zhou et al., 2023) for the "precise instruction following" domain. While the domain types for the training and test sets were the same five domains listed above, the specific tasks used within each domain were configured differently. Thus, all tasks in the test set were unseen. As shown in Table F8, the experimental results show that our method outperforms other merging baselines in mathematics reasoning and precise instruction following. Specifically, in mathematics reasoning, our approach surpasses not only the merging baselines but also the performance of the single fine-tuned model. Although our score in general instruction and knowledge recall following is slightly lower than TIES, which achieves the top performance, the difference is negligible. This demonstrates that our method functions well for LLMs and exhibits excellent generalization capability, even on unseen tasks.

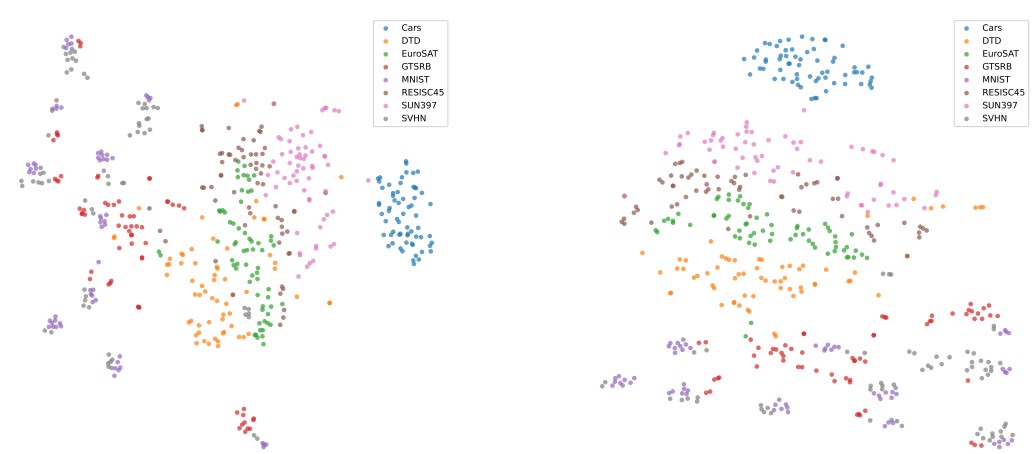

Figure F4: **Task-specific feature distributions from the merged model, extracted via Task Arithmetic (left) and TIES-Merging (right) approximated as Gaussian distributions for 8 tasks.**

Table F8: **Performance of Llama3.1-8B across the four NLP domains.** Math refers to mathematics reasoning, General IF refers to general instruction following, Precise IF refers to precise instruction following, and Knowledge refers to knowledge recall. Bold indicates the best performance excluding the fine-tuned model.

| Method | Math | General IF | Precise IF | Knowledge | Avg. |
|---|---|---|---|---|---|
| Fine-tuned | 0.528 | 0.650 | 0.715 | 0.295 | 0.547 |
| Weight Averaging | 0.467 | **0.663** | 0.577 | **0.319** | 0.507 |
| Task Arithmetic | 0.468 | 0.662 | 0.584 | **0.319** | 0.508 |
| TIES-Merging | 0.528 | 0.655 | 0.610 | 0.313 | 0.527 |
| **EMR + MAD (Ours)** | **0.552** | 0.640 | **0.656** | 0.318 | **0.542** |
| **TM-TA + MAD (Ours)** | 0.546 | 0.636 | 0.608 | 0.318 | 0.527 |

## G DISCLOSURE OF LARGE LANGUAGE MODEL USAGE

In accordance with the ICLR 2026 policy on the use of Large Language Models (LLMs) in paper writing, we disclose that LLMs were utilized to aid or polish the writing of this paper. Specifically, an LLM was employed for minor grammar corrections, improving sentence structure, and refining overall clarity and conciseness of the text. This usage was solely for stylistic and linguistic improvements to the generated content by the human authors and did not involve content generation for the core scientific ideas, experimental design, results, or conclusions. All scientific contributions, intellectual property, and original ideas presented in this work remain those of the human authors.

