# OpenReview forum: "Training-free Task Classification for Multi-Task Model Merging"
_ICLR.cc/2026/Conference — Submitted to ICLR 2026_

### Official Review · Reviewer_Jwgd · 2025-10-16

**Soundness:** 3
**Presentation:** 2
**Contribution:** 2
**Rating:** 4
**Confidence:** 4

**Summary:**

The paper introduces MAD, a training-free framework designed to empower existing subspace-based model merging methods to operate in task-unknown scenarios. It leverages Gaussian Discriminant Analysis by first creating a merged model via simple weight averaging. Using this model, it extracts feature representations for a small sample of training data from each task and models each task's feature distribution as a multivariate Gaussian. At inference time, MAD calculates the Mahalanobis distance between the features of a new input and the pre-computed mean and covariance of each task's distribution. The task corresponding to the minimum distance is selected, and its specific binary mask is used to configure the final merged model for prediction.

**Strengths:**

* This approach provides a clear and creative departure from existing methods that either require training a dedicated routing network (e.g., TWIN-Merging) or rely on computationally expensive multiple forward passes (e.g., DaWin). The "plug-and-play" nature of MAD, allowing it to be combined with various subspace merging methods.
* The proposed method is well-motivated and clearly described. The empirical evaluation is thorough and convincing, spanning multiple domains (vision and NLP). The authors include comprehensive comparisons against relevant baselines. Furthermore, the ablation studies on the choice of distance metric (Table 6) and the number of samples needed for distribution estimation (Table 5) provide strong support for their design choices.
* The requirement of knowing the task ID at inference is a major barrier to deploying these models in real-world systems.

**Weaknesses:**

* The method computes the initial feature distributions using a model created by simple weight averaging (θ_merge). The paper's own results (Tables 2 and 3) confirm that this "Weight Averaging" baseline performs poorly on the end tasks. This creates a conceptual tension: the paper implicitly argues that this poorly performing model produces a feature space with enough class separation to enable highly accurate task classification.
* The experiments are conducted on task sets that are largely distinct (e.g., MNIST digits, EuroSAT satellite images, Cars). The paper does not explore how MAD would perform when faced with a set of very similar, fine-grained tasks (e.g., classifying 20 different species of birds or 15 different models of cars). In such scenarios, the feature distributions would likely have much greater overlap, posing a more significant challenge to the Mahalanobis distance-based classifier.

**Questions:**

Some symbols in the text are not used consistently. For example, in lines 304–305, the merged task vector is sometimes denoted as $\tau_{merge}$ and sometimes as $\Delta_{merge}$.

---

> ### Author Response · Authors · 2025-11-26
>
> We are thankful for the reviewer's careful reading and helpful criticisms, which have allowed us to strengthen our submission considerably.
>
> > Task classification analysis across different merging methods
>
> | Feature Extracting Method | Task classification Avg |
> | --- | --- |
> | Weight Averaging (Ours) | 98.0 |
> | Task Arithmetic (task-unknown) | 92.7 |
> | TIES-Merging (task-unknown) | 90.9 |
>
> We thank the reviewer for highlighting this astute observation regarding a potential conceptual tension between the multi-task performance of Weight Averaging and its utility for task classification. We agree this point warrants a thorough investigation.
>
> To address this, we have conducted an experiment comparing the efficacy of features extracted from different merging methods—namely, Task Arithmetic, TIES-Merging, and our chosen Weight Averaging—for the purpose of task classification. We have used the 'ViT-B/32' model to perform 8-task classification, analyzing the distributions of features extracted by each pre-merged model.
>
> As depicted in Figure F3 (Line 1316 - 1333) and Figure F4 (Line 1350 - 1367) of revision, our analysis of extracted feature distributions reveals that the **Weight Averaging consistently forms the most distinctly separated task clusters, whereas other approaches exhibit noticeable overlaps**. This observed overlap in other merging methods such as Task Arithmetic and TIES-Merging, we hypothesize, complicates global task identification and consequently contributes to their diminished task classification performance. Our empirical results further corroborate this: the **Weight Averaging approach achieved the highest task classification performance, thereby directly justifying its selection** for feature extraction within MaD.

---

> ### Author Response · Authors · 2025-11-26
>
> > Analysis of feature distribution and performance on similar datasets
>
> | Method | Multi-task Avg | Task classification Avg |
> | --- | --- | --- |
> | Fine-tuned | 78.1 | - |
> | Weight Averaging | 60.9 | - |
> | Task Arithmetic (task-unknown) | 51.1 | - |
> | TIES-Merging (task-unknown) | 61.4 | - |
> | DaWin | 61.0 | - |
> | TWIN | 64.6 | 55.9 |
> | MaD (Ours) | 66.9 | 59.4 |
>
> We thank the reviewer for highlighting the critical challenge posed by similar, fine-grained tasks to MaD's performance. We agree that evaluating such scenarios is crucial for assessing its robustness. To address this, we have conducted an experiment using a **birds dataset (CUB-200)** to specifically test MaD's capabilities on fine-grained, highly similar tasks.
>
> To prevent class overlap, we split the CUB-200 dataset into 8 distinct classes, using each as a separate task. We then have fine-tuned parameters for each of these split datasets for the experiment . First, we have examined the feature distributions of the samples used for mean and covariance extraction to assess if distinct clusters could be formed even with similar tasks. Second, we have measured both task classification and multi-task performance to compare MaD against existing methods.
>
> As shown in Figure F2 (Line 1296 - 1314) of revision, while the extracted feature distributions, as expected, do not form clusters as distinctly separated as those from highly distinct tasks, MaD still achieves **better task classification performance compared to TWIN-Merging's trained router**. This suggests that even when feature distributions are less clearly separated, MaD's approach of approximating the unknown true distribution with a Gaussian, in many cases, proves more effective.
>
> Furthermore, when MaD is added to  subspace-based model merging method, it achieves **superior multi-task performance compared to other approaches designed for task-unknown scenarios.** This indicates that MaD can be applied as an add-on to subspace-based model merging across all task-unknown scenarios (whether tasks are distinct or similar), providing both improved task classification and enhanced multi-task performance with  additional training.
>
> > Correcting symbolic errors
>
> We sincerely for the inconsistent use of symbols in the manuscript. We deeply appreciate the reviewer's careful attention to detail in identifying this and other related errors. We have thoroughly reviewed the entire manuscript and ensured that all symbols are now used consistently throughout the paper. Specifically, we have standardized the notation for the merged task vector and corrected any other inconsistencies. We are grateful for helping us enhance the precision and clarity of our notation.

---

### Official Review · Reviewer_5sgK · 2025-10-26

**Soundness:** 2
**Presentation:** 3
**Contribution:** 2
**Rating:** 4
**Confidence:** 4

**Summary:**

This paper addresses the problem of merging task-specific models into a unified multi-task model when the task identity at inference time is unknown. Their contributions include:
.
* They propose to treat identifying the task of a given input as a task classification problem, without requiring extra training or multiple forward-passes through multiple models. Specifically, they use Gaussian discriminant analysis (GDA) and Mahalanobis distance on features extracted from a pre-merged model to estimate which task distribution the input belongs to.


* The authors demonstrate the method on both vision and NLP domains under multi-task merging with unknown task IDs. They show that MAD improves performance (compared to baselines that either rely on training a router or multiple forward-passes) while maintaining training‐free and single pass inference complexity.


* Compared to other “task‐unknown” merging methods that require multiple forward passes or a trained router network, MAD claims to provide a plug‐and‐play, efficient alternative (single forward pass, no extra training) with lower memory overhead (due to binary masks rather than full expert models)

**Strengths:**

* The problem of task unknown at inference is realistic (in deployment scenarios where an input might come from any of many tasks) and under‐addressed in merging literature. The framing is compelling.


* The claim of requiring no additional training and only one forward pass per input (rather than multiple experts) is a strong operational advantage which enhances practicality.


* The empirical demonstration in both vision and NLP domains strengthens the argument that the method is general rather than domain‐specific.

**Weaknesses:**

The method assumes that the merged model’s feature distributions for each task can be approximated by a multivariate Gaussian (mean µₜ, covariance Σₜ). While the authors provide some empirical justification (Figure 3) showing clustering/separation of features, it remains a strong assumption, especially as tasks scale or become more heterogeneous.


* It is unclear how well the method scales when there are many (e.g., dozens or hundreds) of tasks, or when tasks are very similar/broadly overlapping. The generality to large task sets needs more evidence.


* Since the method uses extracted features and covariance estimates from the merged model, any drift in feature representation (e.g., as new tasks are added, or distribution shifts) may degrade task classification. How robust is the method to such changes?


* The approach assumes that the pre‐merged model yields meaningful features for task separation. If the merging is poor or tasks are interfering heavily, then the feature separation may collapse and the Mahalanobis classification may fail. The method seems to build on top of “good” merging methods rather than solving merging itself.



* While Table 1 and other sections compare to other task‐unknown merging methods, it could be helpful to include classical routing approaches (trained router), open‐set classification approaches, or evaluate cost/latency vs. inference performance.

**Questions:**

* Have you evaluated the method when T is large (e.g., > 20) or when tasks cover very diverse domains? What is the decline in accuracy or task‐classification error when T increases?


* You estimate the task‐conditional feature covariance matrix Σₜ using N = 64 samples by default . How sensitive is the method to the number of samples, and to the estimation quality (especially for high dimensional features)? How do you ensure Σₜ is well‐conditioned?


* How does your method behave when the input comes from a task that was not in the training set of tasks (i.e., an entirely new task)? Does the Mahalanobis distance raise a flag (e.g., high distance for all tasks) or will it mis‐classify to the “closest” known task? Have you considered detection of unknown tasks?


* Could you provide detailed latency and memory cost comparisons (single forward pass with mask vs. multiple passes or trained router) in your experiments? This would help quantify the practical advantage.

---

> ### Author Response · Authors · 2025-11-26
>
> We extend our sincere appreciation to the reviewer for their time and effort in providing such comprehensive feedback.
>
> > Further examination of feature Gaussian distribution
>
> We appreciate the reviewer's insightful comment regarding the foundational assumption of Gaussian feature distributions, especially the concern about potential multi-modal distributions. To rigorously validate the effectiveness of modeling features as Gaussian in this context, we have conducted a comprehensive comparative analysis of task classification performance across different routing approaches.
>
> Specifically, we compare our MaD method (which assumes and fits features to a Gaussian distribution) against k-NN (k-nearest neighbors, a non-parametric method making no explicit distributional assumptions) and TWIN-Merging (which utilizes an MLP-based learned router to fit features). MaD and k-NN are both training-free, each utilizing only 64 samples per task based on Mahalanobis and cosine distances, respectively. By contrast, TWIN-Merging employs a **learned router** and requires a significantly more samples (1000 samples) per task for its task classification. For a more robust verification, these evaluations were performed across various ViT backbone models ({B/32, B/16, L/14}) and different numbers of tasks (8, 14, and 20 tasks).
>
> - Number of tasks = 8
>
> |  | **MaD** | k-NN (k=1) | k-NN (k=3) | k-NN (k=5) | k-NN (k=10) | TWIN |
> | --- | --- | --- | --- | --- | --- | --- |
> | ViT-B/32 | **98.0** | 93.7 | 92.5 | 91.7 | 89.4 | 90.5 |
> | ViT-B/16 | **98.3** | 94.7 | 93.6 | 92.8 | 91.3 | 86.8 |
> | ViT-L/14 | **98.5** | 93.2 | 92.8 | 92.2 | 90.7 | 83.7 |
> - Number of tasks = 14
>
> |  | **MaD** | k-NN (k=1) | k-NN (k=3) | k-NN (k=5) | k-NN (k=10) | TWIN |
> | --- | --- | --- | --- | --- | --- | --- |
> | ViT-B/32 | **96.9**  | 92.9 | 92.0 | 91.2 | 89.1 | 84.0 |
> | ViT-B/16 | **97.0** | 93.4 | 92.4 | 91.7 | 90.2 | 79.9 |
> | ViT-L/14 | **97.7** | 93.7 | 93.5 | 93.2 | 92.0 | 72.6 |
> - Number of tasks = 20
>
> |  | **MaD** | k-NN (k=1) | k-NN (k=3) | k-NN (k=5) | k-NN (k=10) | TWIN |
> | --- | --- | --- | --- | --- | --- | --- |
> | ViT-B/32 | **94.1** | 90.0 | 89.5 | 89.0 | 86.8 | 75.8 |
> | ViT-B/16 | **94.7** | 90.4 | 89.4 | 88.8 | 87.0 | 77.6 |
> | ViT-L/14 | **95.8** | 91.3 | 91.2 | 91.3 | 90.2 | 78.8 |
>
> Our results demonstrate that MaD consistently achieves strong task classification performance across diverse settings. **This indicates that, in scenarios where the true feature distribution is unknown, approximating it with a Gaussian distribution (as done in MaD) proves more effective than either making no explicit distributional assumption (k-NN) or utilizing a data-intensive learned router (TWIN-Merging).** Notably, MaD consistently surpasses TWIN-Merging in task classification performance, despite TWIN-Merging employing a learned router trained with substantially more examples (i.e., 1000 samples per task), while MaD is a training-free approach requiring significantly fewer samples (i.e., 64 samples per task) for modeling its Gaussian distributions.
>
> While multi-modal Gaussian distributions could theoretically be modeled more accurately using approaches like Gaussian Mixture Models (GMM), such methods would incur significant practical drawbacks. Specifically, **GMMs would require storing means and covariances for each mode, leading to a linear increase in storage and a corresponding rise in inference cost with the number of modes.** However, our current approach, which simplifies by fitting a single Gaussian, already yields sufficiently strong performance. This pragmatic balance of accuracy and efficiency positions MaD as an effective and practical methodology for real-world applications.

---

> ### Author Response · Authors · 2025-11-26
>
> > Scalability to large and diverse task sets.
>
> - Task classification Avg performance (ViT-B/32)
>
> | Method | 30 tasks | 50 tasks |
> | --- | --- | --- |
> | **MaD** | **93.6** | **85.9** |
> | k-NN (k=1) | 89.2 | 80.9 |
> | k-NN (k=3) | 88.9 | 80.4 |
> | k-NN (k=5) | 88.6 | 80.3 |
> | k-NN (k=10) | 87.1 | 79.1 |
> | TWIN | 63.1 | 45.3 |
> - Multi-task Avg performance (ViT-B/32)
>
> | Method | 30 tasks | 50 tasks |
> | --- | --- | --- |
> | Fine-tuned | 93.1 | 90.8 |
> | Weight Averaging | 59.1 | 43.8 |
> | Task Arithmetic (task-unknown) | 17.5 | 10.8 |
> | TIES-Merging (task-unknown) | 56.1 | 42.7 |
> | TWIN-Merging | 60.1 | 16.4 |
> | DaWin | 40.3 | 33.1 |
> | WEMoE | 67.1 | 51.6 |
> | MoW-Merging | 56.4 | 30.1 |
> | **EMR+MaD (Ours)** | **88.1** | **78.6** |
> | **TM-TA + MaD (Ours)** | **88.5** | **80.5** |
> | **TM-TIES + MaD (Ours)** | **89.6** | **82.5** |
> - Accuracy on similar datasets (CUB-200)
>
> | Method | Multi-task Avg | Task classification Avg |
> | --- | --- | --- |
> | Fine-tuned | 78.1 | - |
> | Weight Averaging | 60.9 | - |
> | Task Arithmetic (task-unknown) | 51.1 | - |
> | TIES-Merging (task-unknown) | 61.4 | - |
> | DaWin | 61.0 | - |
> | TWIN | 64.6 | 55.9 |
> | **MaD (Ours)** | **66.9** | **59.4** |
>
> We thank the reviewer for raising this crucial point about scalability to a larger number of tasks and robustness to task similarity. To address this, we have extended our evaluation beyond 20 tasks to scenarios involving **30 and 50 tasks**, investigating both feature distribution and task classification performance.
>
> For these expanded task sets, we have utilized the 'ViT-B/32' model and additionally compared MaD task classification performance against k-NN (making no explicit distributional assumptions) and TWIN-Merging (which uses an MLP-based learned router).
>
> As shown in Figure F1 (Line 1242 - 1259) of revision, our experimental results reveal that even with a larger number of tasks (up to 50), although some overlaps are observed where domains inherently share similarities, the majority of task clusters remain well separated. Importantly, task-classification performance does not experience significant degradation as the number of tasks increases to 30 and 50, and MaD consistently outperforms both k-NN and the TWIN Router in these larger-scale settings.
>
> Furthermore, we show the additional results on **30-task and 50-task multitask performance**. Our experimental results consistently show that MaD with existing merging methods maintain the **best performance even as the number of tasks increases**. This further underscores MaD's superior scalability and efficiency in high-task environments.
>
> In addition, we have conducted an additional experiment on highly similar datasets using only the CUB-200 birds dataset detailed in Appendix F: Performance on Fine-Grained and Similar Tasks (Line 1131 - 1229) of revision and `Jwgd`’s W2 response. As shown in Figure F2 (Line 1296 - 1314) of revision, even in this scenario, where feature overlaps are inherently more pronounced, we observe that MaD (best performance among EMR+MaD, TM-TA+MAD and TM-TIES+MaD) exhibited superior multi-task performance and task classification performance compared to other methods. This further confirms MaD's robustness and strong generalization capabilities across diverse as well as fine-grained task sets.

---

> ### Author Response · Authors · 2025-11-26
>
> > Analysis of robustness to unseen task
>
> | Method | Seen avg (8 tasks) | Unseen avg (6 tasks) | Avg. performance |
> | --- | --- | --- | --- |
> | Task Arithmetic (task-unknown) | 61.3 | 44.3 | 54.0 |
> | TIES-Merging (task-unknown) | 74.0 | 55.2 | 65.9 |
> | Dawin | 89.0 | 57.5 | 75.5 |
> | Twin | 83.4 | 62.6 | 74.5 |
> | **EMR+MaD (Ours)** | **90.5** | **57.8** | **76.5** |
> | **TM-TA+MaD (Ours)** | **92.1** | **63.8** | **78.0** |
> | **TM-TIES+MaD (Ours)** | **91.9** | **64.4** | **80.1** |
>
> |  | Seen | Seen | Seen | Seen | Seen | Seen | Seen | Seen | Unseen | Unseen | Unseen | Unseen | Unseen | Unseen |
> | --- | --- | --- | --- | --- | --- | --- | --- | --- | --- | --- | --- | --- | --- | --- |
> | Task | Cars | DTD | EuroSAT | GTSRB | MNIST | RESISC45 | SUN397 | SVHN | CIFAR100 | STL10 | Flowers102 | OxfordIIITPet | PCAM | FER2013 |
> | Entropy  | 0.002 | 0.0123 | 0.0045 | 0.0099 | 0.0004 | 0.0228 | 0.0624 | 0.0092 | 0.4498 | 0.3632 | 0.0030 | 0.3275 | 0.4017 | 0.7141 |
>
> We appreciate the reviewer's comment regarding the robustness of MaD to feature distribution shift. To thoroughly validate MaD's robustness in scenarios where new, unseen tasks are encountered, we have designed an experiment to simulate such conditions.
>
> Specifically, we have designated tasks used for 8-task classification as "seen tasks." For "unseen tasks," we have utilized the additional 6 tasks from our 14-task classification dataset that were not part of the 8-task set. We then have measured the performance of MaD when confronted with these entirely new task distributions.
>
> Our results demonstrate that when MaD is integrated with existing subspace-based model merging methods, it achieves **superior performance on these unseen tasks compared to other task-unknown scenario baselines like TWIN-Merging and DaWin**. This outcome strongly indicates that MaD maintains robust task classification capability even for entirely unknown tasks, rather than simply misclassifying them.
>
> Furthermore, regarding the detection of unknown tasks: we have measured the entropy of features from both seen and unseen tasks using a validation set. Our analysis reveals that for all but one task, the **entropy for unseen tasks are markedly higher than that for seen tasks**. This suggests that the "high distance for all tasks" scenario the reviewer mentioned can indeed be observed through entropy values. Consequently, these entropy values could potentially be leveraged as a "flag" to differentiate between seen and unseen tasks.
>
> > Task-classification analysis under various merging methods
>
> | Feature Extracting Method | Task classification  Avg |
> | --- | --- |
> | Weight Averaging (Ours) | 98.0 |
> | Task Arithmetic (task-unknown) | 92.7 |
> | TIES-Merging (task-unknown) | 90.9 |
>
> We thank the reviewer for this insightful comment, which critically examines our method's reliance on the quality of features from the pre-merged model. To directly investigate whether a "good" merging method, in terms of overall multi-task performance, necessarily produces features well-suited for task separation, we have conducted an experiment comparing task classification performance using features extracted from models pre-merged with different strategies.
>
> Specifically, we have utilized the 'ViT-B/32' model and evaluated 8 tasks, extracting features from models pre-merged using Task Arithmetic, TIES-Merging, and our chosen Weight Averaging method.
>
> As shown in Figure F3 (Line 1316 - 1333) and Figure F4 (Line 1350 - 1367) of revision, when examining the feature distributions extracted by each methodology, we observe that the **Weight Averaging merging method consistently formed the most distinctly separated task clusters**, while other merging approaches exhibited some overlaps. We hypothesize that **the observed overlapping clusters in other merging approaches complicate global task identification, which likely contributes to the slight degradation in their task classification performance.** Indeed, our empirical task classification results confirm that the Weight Averaging approach yielded the highest performance, thus **directly justifying our choice** of this method for feature extraction within MaD.

---

> ### Author Response · Authors · 2025-11-26
>
> > Extended comparison to task-unknown merging methods
>
> - Multi-task Avg performance
>
> | Method | ViT-B/32 8 tasks | ViT-B/32 14 tasks | ViT-B/32 20 tasks | ViT-B/16 8 tasks | ViT-B/16 14 tasks | ViT-B/16 20 tasks | ViT-L/14 8 tasks | ViT-L/14 14 tasks | ViT-L/14 20 tasks |
> | --- | --- | --- | --- | --- | --- | --- | --- | --- | --- |
> | Fine-tuned | 92.8 | 90.9 | 91.3 | 94.7 | 92.8 | 92.8 | 95.9 | 94.3 | 94.8 |
> | Weight Averaging | 66.3 | 64.3 | 61.0 | 72.2 | 69.5 | 65.3 | 79.6 | 76.7 | 71.6 |
> | Task Arithmetic (task-known) | 70.8 | 65.3 | 60.5 | 75.4 | 70.5 | 65.8 | 84.9 | 79.4 | 74.0 |
> | TIES-Merging (task-known) | 75.1 | 68.0 | 63.4 | 79.7 | 73.2 | 68.2 | 86.9 | 79.5 | 75.7 |
> | Task-unknown scenarios |  |  |  |  |  |  |  |  |  |
> | Task Arithmetic (task-unknown) | 61.3 | 38.2 | 23.6 | 68.8 | 45.5 | 25.6 | 83.4 | 66.6 | 42.7 |
> | TIES-Merging (task-unknown) | 74.0 | 65.0 | 58.2 | 79.6 | 71.3 | 65.8 | 86.9 | 78.7 | 74.4 |
> | TWIN-Merging | 84.0 | 70.0 | 57.5 | 91.4 | 78.4 | 63.1 | 93.7 | 86.2 | 74.8 |
> | DaWin | 89.0 | 73.8 | 52.8 | 87.1 | 77.8 | 62.8 | 91.6 | 82.6 | 77.5 |
> | WEMoE | 90.4 | 83.1 | 74.4 | 93.1 | 84.0 | 76.6 | 94.8 | 87.0 | 75.7 |
> | MoW-Merging | 88.1 | 83.2 | 79.3 | 93.7 | 79.3 | 78.2 | 94.9 | 78.8 | 81.8 |
> | **EMR+MaD (Ours)** | **90.4** | **86.8** | **85.6** | **92.8** | **89.8** | **88.7** | **95.0** | **92.5** | **92.0** |
> | **TM-TA+MaD (Ours)** | **92.0** | **89.4** | **88.7** | **93.9** | **91.0** | **91.1** | **92.1** | **89.0** | **89.8** |
> | **TM-TIES+MaD (Ours)** | **91.9** | **89.1** | **88.2** | **93.8** | **91.4** | **90.7** | **93.9** | **90.3** | **90.6** |
> - Inference cost
>
> | Method | Batch inference | Inference cost | Memory (VRAM) | Avg. performance |
> | --- | --- | --- | --- | --- |
> | Task Arithmetic  (task-unknown) | o | 0.0008s | 1.0GB | 61.3 |
> | TIES-Merging  (task-unknown) | o | 0.0008s | 1.0GB | 74.0 |
> | TWIN-Merging | x | 0.03s | 5.6GB | 84.0 |
> | DaWin w/o mixture modeling | x | 0.63s | 5.5GB | 84.2 |
> | DaWin | o | 0.001s | 9.2GB | 89.0 |
> | WEMoE | x | 0.02s | 4.3GB | 90.4 |
> | MoW-Merging | o | 0.008s | 4.9GB | 88.1 |
> | **EMR+MaD (Ours)** | o| **0.002s** | **1.7GB** | **90.4** |
> | **TM-TA+MaD (Ours)** | o | **0.004s** | **1.1GB** | **92.0** |
> | **TM-TIES+MaD (Ours)** | o| **0.004s** | **1.1GB** | **91.9** |
>
> We thank the reviewer for this valuable suggestion to broaden our comparison with trained router approaches. We agree that such an expanded evaluation would further strengthen our claims.
>
> Beyond the existing comparison with dynamic approaches that utilize multiple forward passes (e.g., DaWin) and employ trained routers (e.g., TWIN-Merging), we have conducted additional experiments under identical settings to compare our method (MaD) against other prominent trained router approaches, specifically **WEMoE and MoW-Merging**. Our results demonstrate that MaD consistently **maintains its advantage in both performance and inference cost** when benchmarked against these powerful dynamic model merging methods. This further corroborates that MaD is highly competitive and robust, even when compared with strong, established trained router techniques. We will integrate these new experimental results and a detailed discussion in revision.
>
> We sincerely appreciate this insightful and constructive comment, as it has undeniably contributed to elevating the quality and robustness of our paper.

---

> ### Author Response · Authors · 2025-11-26
>
> > Comparison of various sample sizes
> >
>
> | Method | 8 tasks | 14 tasks | 20 tasks |
> | --- | --- | --- | --- |
> | MaD $(N=4)$ | 90.5 | 89.9 | 80.5 |
> | MaD $(N=8)$ | 93.5 | 91.9 | 87.4 |
> | MaD $(N=N_{train})$ | 98.6 | 97.5 | 96.2 |
> | MaD $(N=64)$ | 98.0 | 96.9 | 94.1 |
>
> We thank the reviewer for these pertinent questions regarding the sensitivity to sample size.
>
> Regarding sensitivity to the number of samples: Our experiments indicate that while task classification performance does show some decline with a reduction in sample size, the method is not overly sensitive. Notably, even when utilizing **as few as 4 samples per task, MaD still achieves over 90% accuracy** for 8-task classification, demonstrating robust performance even under severe data scarcity.

---

### Official Review · Reviewer_75xv · 2025-10-28

**Soundness:** 2
**Presentation:** 2
**Contribution:** 2
**Rating:** 2
**Confidence:** 4

**Summary:**

This paper proposes a model-merging method for task-unknown scenarios that assembles the merged model by automatically identifying the task id to which each test sample belongs at inference time. However, the proposed approach incurs additional inference-time overhead, lacks validation on large-scale models, and contains numerous grammatical and spelling issues.

**Strengths:**

- This paper improves the accuracy of the merged model by distinguishing task IDs through the extraction of the mean and covariance matrices of each task’s data.
- The overall structure and organization of the paper are well-designed.

**Weaknesses:**

- The proposed method is straightforward, offering limited technical contribution.
- The approach relies on extracting the mean and covariance of the data. However, most model merging scenarios assume that no data is available.
- The MAD method requires each sample to undergo a forward pass before actual inference, which reduces inference efficiency. In Table 4, the inference cost is not compared with static merging methods such as Task Arithmetic or Ties-Merging.
- The definition of the task-unknown scenario in this paper is inaccurate. For instance, methods like Task Arithmetic and Ties-Merging do not require task IDs, while dynamic merging approaches seem more consistent with the scenario described in this paper.
- The paper lacks comparison with dynamic merging approaches such as WEMoE [1] and MoW-Merging [2], which are conceptually similar to Twin-Merging.
- There is no validation of the proposed method on larger-scale language models.
- The paper contains numerous typographical and grammatical errors, including but not limited to:
  - Line 23: “belong sto” -> “belong to”
  - Line 48: Incomplete usage of “either”
  - Line 66: Incomplete “not only …” structure
  - Line 67: “multiple forward pass” -> “multiple forward passes”
  - Line 78: The sentence “Consequently, the input is ultimately predicted by the finally merged model.” is inaccurate—why would the input be predicted?
  - Line 450: Redundant phrase in “Gaussian distribution’s covariance (third row) (fourth row)” (the first parenthetical note should be removed).

[1]Tang, Anke, et al. "Merging multi-task models via weight-ensembling mixture of experts." Proceedings of the 41st International Conference on Machine Learning. 2024.

[2]Ye, Peng, et al. "Dynamic model merging with mixture of weights." IEEE Transactions on Circuits and Systems for Video Technology (2025).

**Questions:**

Refer to the Weaknesses section

---

> ### Author Response · Authors · 2025-11-26
>
> We are grateful for the reviewer's constructive comments, which have significantly helped us improve the clarity and quality of our work.
>
> > Clarification of the task-unknown scenario relative to existing methods
> >
> - Multi-task Avg performance
>
>
> | Method | ViT-B/32 8 tasks | ViT-B/32 14 tasks | ViT-B/32 20 tasks | ViT-B/16 8 tasks | ViT-B/16 14 tasks | ViT-B/16 20 tasks | ViT-L/14 8 tasks | ViT-L/14 14 tasks | ViT-L/14 20 tasks |
> | --- | --- | --- | --- | --- | --- | --- | --- | --- | --- |
> | Fine-tuned | 92.8 | 90.9 | 91.3 | 94.7 | 92.8 | 92.8 | 95.9 | 94.3 | 94.8 |
> | Weight Averaging | 66.3 | 64.3 | 61.0 | 72.2 | 69.5 | 65.3 | 79.6 | 76.7 | 71.6 |
> | Task Arithmetic (task-known) | 70.8 | 65.3 | 60.5 | 75.4 | 70.5 | 65.8 | 84.9 | 79.4 | 74.0 |
> | TIES-Merging (task-known) | 75.1 | 68.0 | 63.4 | 79.7 | 73.2 | 68.2 | 86.9 | 79.5 | 75.7 |
> | Task-unknown scenarios |  |  |  |  |  |  |  |  |  |
> | Task Arithmetic (task-unknown) | 61.3 | 38.2 | 23.6 | 68.8 | 45.5 | 25.6 | 83.4 | 66.6 | 42.7 |
> | TIES-Merging (task-unknown) | 74.0 | 65.0 | 58.2 | 79.6 | 71.3 | 65.8 | 86.9 | 78.7 | 74.4 |
> | TWIN-Merging | 84.0 | 70.0 | 57.5 | 91.4 | 78.4 | 63.1 | 93.7 | 86.2 | 74.8 |
> | DaWin | 89.0 | 73.8 | 52.8 | 87.1 | 77.8 | 62.8 | 91.6 | 82.6 | 77.5 |
> | EMR+MaD (Ours) | 90.4 | 86.8 | 85.6 | 92.8 | 89.8 | 88.7 | 95.0 | 92.5 | 92.0 |
> | TM-TA+MaD (Ours) | 92.0 | 89.4 | 88.7 | 93.9 | 91.0 | 91.1 | 92.1 | 89.0 | 89.8 |
> | TM-TIES+MaD (Ours) | 91.9 | 89.1 | 88.2 | 93.8 | 91.4 | 90.7 | 93.9 | 90.3 | 90.6 |
>
> We thank the reviewer for comments on the operational context of Task Arithmetic and TIES-Merging. While these methods are formulated for task-unknown scenarios, the reported best results in their respective papers are obtained under task-known scenarios. **To elaborate, both Task Arithmetic and TIES-Merging** **require tuning optimal scaling coefficients ($\lambda_t$) for each individual task, typically via a validation set (As detailed in the section C.4 of [1]) . During inference, they apply these task-specific optimal ($λ_t$
> ) values, thereby inherently assuming prior knowledge of the task.**
>
> **To facilitate a fair comparison in task-unknown scenarios, in additional experiments shown in the above table, we have adapted these methods by adopting a fixed hyperparameter setting for all tasks, following the configurations suggested in the TIES-Merging paper.** Specifically, we have applied $λ=0.4$
>  for Task Arithmetic and $K=20, \lambda=1$ for TIES-Merging across all tasks.
>
> Our experimental results demonstrate that under these task-unknown conditions, both Task Arithmetic and TIES-Merging exhibit significantly degraded performance compared to other model merging methods, including MaD, designed for such task-unknown scenarios.
>
> To ensure clearer distinction and prevent any potential confusion, we will add these performance results for Task Arithmetic and TIES-Merging under task-unknown scenarios along with a more precise discussion of their operational assumptions.
>
> > Contribution of MaD
> >
>
> While our approach may appear straightforward, we believe it offers several distinct and impactful contributions, which have also been highlighted by other reviewers
>
> 1. As Reviewers `2yxj`, `5sgK`, and `Jwgd` have noted, enabling model merging in **task-unknown scenarios** is a significant and challenging research direction. Our method effectively achieves this for diverse models without substantial performance degradation.
> 2. Reviewers `2yxj` and `Jwgd` pointed out that our method demonstrates **strong versatility** as it can be easily added as an add-on to various existing model merging techniques.
> 3. Unlike conventional methods that require the training of a feature extractor or classification head with hundreds to thousands of data samples per task, our approach achieves accurate task classification with **only few tens of samples per task and without the need for separate parameter optimization (training).** This significantly reduces the cost associated with model training when new tasks are added or existing tasks are removed, thus demonstrating high efficiency and practical utility.
> 4. We show that it is effective to employ Mahalanobis distance **for task classification**, which is a distinct and valuable application within the context of model merging.

---

> ### Author Response · Authors · 2025-11-26
>
> > Availability of data in merging scenarios.
> >
>
> We thank the reviewer for raising this pertinent concern regarding data availability. While it is true that some model merging scenarios assume data is entirely unavailable, many prominent existing methods also require access to data for their operation. For instance, Task Arithmetic [2] and TIES-Merging [1] utilize validation sets for hyperparameter tuning of scaling coefficients, while AdaMerging [3] and similar approaches learn scaling coefficients directly from test sets. Furthermore, methods like WEMoE [4], TWIN-Merging [5] and MoW-Merging [6] explicitly necessitate thousands of samples per task for router training.
>
> By contrast, our method demonstrates remarkable data efficiency. It requires **only few tens of samples per task from the training set (instead of using extra data from validation or test in Task Arithmetic, TIES-Merging and AdaMerging)** and yet consistently achieves high performance across both computer vision and natural language processing domains. Moreover, our approach can be considered efficient because, much like how Batch Normalization layers learn the mean and covariance of activations without storing the raw data itself, our method similarly **stores only the pre-computed mean and covariance for each task's features**, rather than retaining the raw task-specific datasets. This allows for a significant reduction in data storage and processing overhead compared to other data-intensive methods.
>
> > Inference efficiency comparison with static merging methods
> >
> - Inference cost
>
> | Method | Batch inference | Inference cost | Memory (VRAM) | Avg. performance |
> | --- | --- | --- | --- | --- |
> | Task Arithmetic (task-unknown) | o | 0.0008s | 1.0GB | 61.3 |
> | TIES-Merging (task-unknown) | o | 0.0008s | 1.0GB | 74.0 |
> | TWIN-Merging | x | 0.03s | 5.6GB | 84.0 |
> | DaWin w/o mixture modeling | x | 0.63s | 5.5GB | 84.2 |
> | DaWin | o | 0.001s | 9.2GB | 89.0 |
> | WEMoE | x | 0.02s | 4.3GB | 90.4 |
> | MoW-Merging | o | 0.008s | 4.9GB | 88.1 |
> | EMR+MaD (Ours) | o | 0.002s | 1.7GB | 90.4 |
> | TM-TA+MaD (Ours) | o | 0.004s | 1.1GB | 92.0 |
> | TM-TIES+MaD (Ours) | o | 0.004s | 1.1GB | 91.9 |
>
> We appreciate the reviewer's observation regarding the inference cost and the comparison with other merging methods. **As detailed in our preceding response**, Task Arithmetic [2] and TIES-Merging [1] are fundamentally designed for task-known scenarios, and even when their hyperparameters are fixed to adapt them for task-unknown scenarios, they exhibit substantial performance degradation." Our primary objective is to address this limitation by improving the performance of static subspace-based methods under **task-unknown scenarios**.
>
> Consequently, the most crucial and relevant comparison for our methodology is against **other dynamic model merging methods** that are also designed to operate in task-unknown scenarios. Within this specific and pertinent class of methods, our approach consistently demonstrates **substantially faster inference, significantly higher memory efficiency, and superior overall performance** compared to other dynamic (task-unknown scenario) approaches.

---

> ### Author Response · Authors · 2025-11-26
>
> > Comparison with other dynamic merging approaches
> >
> - Multi-task Avg performance
>
> | Method | ViT-B/32 8 tasks | ViT-B/32 14 tasks | ViT-B/32 20 tasks | ViT-B/16 8 tasks | ViT-B/16 14 tasks | ViT-B/16 20 tasks | ViT-L/14 8 tasks | ViT-L/14 14 tasks | ViT-L/14 20 tasks |
> | --- | --- | --- | --- | --- | --- | --- | --- | --- | --- |
> | Fine-tuned | 92.8 | 90.9 | 91.3 | 94.7 | 92.8 | 92.8 | 95.9 | 94.3 | 94.8 |
> | Weight Averaging | 66.3 | 64.3 | 61.0 | 72.2 | 69.5 | 65.3 | 79.6 | 76.7 | 71.6 |
> | Task Arithmetic (task-known) | 70.8 | 65.3 | 60.5 | 75.4 | 70.5 | 65.8 | 84.9 | 79.4 | 74.0 |
> | TIES-Merging (task-known) | 75.1 | 68.0 | 63.4 | 79.7 | 73.2 | 68.2 | 86.9 | 79.5 | 75.7 |
> | Task-unknown scenarios |  |  |  |  |  |  |  |  |  |
> | TWIN-Merging | 84.0 | 70.0 | 57.5 | 91.4 | 78.4 | 63.1 | 93.7 | 86.2 | 74.8 |
> | DaWin | 89.0 | 73.8 | 52.8 | 87.1 | 77.8 | 62.8 | 91.6 | 82.6 | 77.5 |
> | WEMoE | 90.4 | 83.1 | 74.4 | 93.1 | 84.0 | 76.6 | 94.8 | 87.0 | 75.7 |
> | MoW-Merging | 88.1 | 83.2 | 79.3 | 93.7 | 79.3 | 78.2 | 94.9 | 78.8 | 81.8 |
> | EMR+MaD (Ours) | 90.4 | 86.8 | 85.6 | 92.8 | 89.8 | 88.7 | 95.0 | 92.5 | 92.0 |
> | TM-TA+MaD (Ours) | 92.0 | 89.4 | 88.7 | 93.9 | 91.0 | 91.1 | 92.1 | 89.0 | 89.8 |
> | TM-TIES+MaD (Ours) | 91.9 | 89.1 | 88.2 | 93.8 | 91.4 | 90.7 | 93.9 | 90.3 | 90.6 |
>
> We are grateful for constructive comments on the comparison with dynamic merging approaches such as WEMoE [4] and MoW-Merging [6] . Following the reviewer’s suggestion, we additionally have conducted experiments under the same evaluation settings as in the main paper with both WEMoE and MoW-Merging as baselines.
>
> In the case of MoW-Merging, we have adopted its best-performing configuration reported in the original paper, i.e., *AdaMerging++ w/ MoW-Merging* using TrivialAugment for data augmentation and with *m = 10*, where *m* denotes the total number of attention and MLP layers to which MoW-Merging is applied.
>
> We have revised the description of these dynamic merging methods in the Related Work section for better clarification. We believe that this more rigorous comparison, prompted by the reviewer’s insightful and constructive comment, has significantly elevated the quality and robustness of our manuscript.
>
> The results show that **MaD consistently outperforms both methods in terms of accuracy (and further maintains an advantage in inference cost).** This demonstrates that MaD remains competitive even against strong and conceptually similar dynamic model merging approaches. We have included new results and discussion in revision. We are grateful for the comments that help us greatly improve the quality of the paper.
>
> > Experiments with Large Language Models
> >
>
> To validate our method on a larger language model, we have additionally employed the LLaMA3.1-8B model. For this setup, the training tasks included [7]: Tulu 3 Persona MATH, OpenMathInstruct 2, and MuminaMath-TIR for the "mathematics reasoning" domain; WildChat, OpenAssistant, and No Robots for the "general instruction following" domain; FLAN v2, SciRIFF, and TableGPT for the "knowledge recall" domain; and Tulu 3 Persona IF for the "precise instruction following" domain.
>
> On the other hand, the test tasks utilized have been GSM8K [8] and MATH [9] for the "mathematics reasoning" domain; BBH (CoT) [10] and DROP [11] for the "general instruction following" domain; PopQA [12] for the "knowledge recall" domain; and IFEval [13] for the "precise instruction following" domain. While the domain types for the training and test sets were the same five domains listed above, the specific tasks used within each domain were configured differently. Thus, all tasks in the test set were unseen.
>
> | Model/Domain | Math | General IF | Precise IF | Knowledge | Avg |
> | --- | --- | --- | --- | --- | --- |
> | Fine-tuned | 0.528 | 0.650 | 0.715 | 0.295 | 0.547 |
> | Weight Averaging | 0.467 | 0.663 | 0.577 | 0.319 | 0.507 |
> | Task Arithmetic | 0.468 | 0.662 | 0.584 | 0.319 | 0.508 |
> | TIES-Merging | 0.528 | 0.655 | 0.610 | 0.313 | 0.527 |
> | EMR + MaD (Ours) | 0.552 | 0.640 | 0.656 | 0.318 | 0.542 |
> | TM-TA + MaD (Ours) | 0.546 | 0.636 | 0.608 | 0.318 | 0.527 |
>
> **The experimental results show that our method outperforms other merging baselines in “mathematics reasoning” and “precise instruction following”.** Specifically, in “mathematics reasoning”, our approach surpasses not only the merging baselines but also the performance of the single fine-tuned model. Although our score in “general instruction following” and “knowledge recall” is slightly lower than Weight Averaging, which achieves the top performance, the difference is negligible. **This indicates that our method functions well for LLMs and demonstrates excellent generalization ability even on unseen tasks.**

---

> ### Author Response · Authors · 2025-11-26
>
> > Fixing typographical and grammatical errors.
> >
>
> We thank the reviewer for pointing out the typographical and grammatical errors present in the submission. We deeply appreciate the reviewer's meticulous attention to detail in identifying these issues. We have thoroughly proofread the entire manuscript and corrected all the errors mentioned by the reviewer, along with other additional errors we identified during our review. We are grateful for helping us improve the quality of our writing.
>
> [1] Yadav et al., Ties-merging: Resolving interference when merging models. In NeurIPS, 2023.
>
> [2] Ilharco et al., Editing Models with Task Arithmetic, ICLR 2023.
>
> [3] Yang et al., AdaMerging: Adaptive Model Merging for Multi-Task Learning, ICLR, 2024.
>
> [4] Tang et al., Merging Multi-Task Models via Weight-Ensembling Mixture of Experts, ICML, 2024.
>
> [5] Lu et al., Twin-Merging: Dynamic Integration of Modular Expertise in Model Merging, NeurIPS, 2024.
>
> [6] Ye et al., Dynamic model merging with mixture of weights IEEE Transactions on Circuits and Systems for Video Technology, 2025.
>
> [7] Lambert et al., Tulu 3: Pushing Frontiers in Open Language Model Post-Training, arXiv, 2024.
>
> [8] Cobbe et al., Training Verifiers to Solve Math Word Problems, arXiv, 2021.
>
> [9] Hendrycks et al., Measuring Mathematical Problem Solving With the MATH Dataset, NeurIPS, 2021.
>
> [10] Suzgun et al., Challenging BIG-Bench Tasks and Whether Chain-of-Thought Can Solve Them., Findings of ACL, 2023.
>
> [11] Dua et al., DROP: A Reading Comprehension Benchmark Requiring Discrete Reasoning Over Paragraphs, NAACL, 2019.
>
> [12] Mallen et al., When Not to Trust Language Models: Investigating Effectiveness of Parametric and Non-Parametric Memories, ACL, 2023.
>
> [13] Zhou et al., Instruction-Following Evaluation for Large Language Models, arXiv, 2023.

---

### Official Review · Reviewer_2yxj · 2025-10-30

**Soundness:** 3
**Presentation:** 3
**Contribution:** 3
**Rating:** 6
**Confidence:** 3

**Summary:**

This paper proposes MAD, a training-free method that enables existing subspace-based model merging techniques to handle "task-unknown" scenarios, where the task identity of an input is unknown during inference. The core idea is to treat the problem as a task classification challenge. MAD uses Gaussian Discriminant Analysis (GDA) to model the feature distribution of a pre-merged model for each task. For a new input, it calculates the Mahalanobis distance between the input's features and each task's distribution, classifying the input to the task with the smallest distance. The corresponding task-specific binary mask is then applied to the pre-merged model for inference. This approach requires no additional training, only a single forward pass, and is memory-efficient, as it stores only a single pre-merged model and binary masks instead of all task-specific models. Experiments on vision and NLP tasks show that MAD outperforms existing task-unknown merging methods while being faster and more memory-efficient.

**Strengths:**

1. The idea of the reframing model merging for task-unknown scenarios as a training-free task classification problem using GDA and Mahalanobis distance seems novel. It offers a simple yet effective plug-and-play solution to a key limitation of existing subspace-based merging methods.

2. The paper is well-supported by extensive experiments across multiple model architectures, task scales, and domains. The results demonstrate clear performance improvements over strong baselines like TWIN-Merging and DaWin. Ablation studies effectively justify key design choices.

3. The paper is generally well-written and logically structured. The problem definition, motivation, method description, and experiments are clearly presented. Figures 1 and 2 help visualize the concept and pipeline effectively.

**Weaknesses:**

1. The core method relies on the assumption that features from the merged model for each task follow a Gaussian distribution. While Figure 3 provides some visual evidence and prior work is cited, a more rigorous quantitative analysis or discussion of the limitations of this assumption across different models and tasks would strengthen the foundation.

2. For models with very high-dimensional feature spaces (large D), estimating and inverting the full covariance matrix can become computationally expensive and numerically unstable. The paper mentions adding a small epsilon to the diagonal but does not deeply discuss the computational overhead or potential limitations for extremely large feature dimensions.

**Questions:**

The experiments are conducted on CLIP-ViT for vision and T5-large for NLP tasks. Is the proposed method generalizable to other model architectures?

---

> ### Author Response · Authors · 2025-11-26
>
> We sincerely thank the reviewer for their thorough review and insightful feedback on our submission.
>
> > Extended analysis of feature Gaussian distribution
>
> We appreciate the reviewer's insightful comment regarding the assumption of Gaussian feature distributions. To validate the effectiveness of modeling features as Gaussian, we have conducted a comprehensive comparative analysis of task classification performance across different routing approaches.
>
> Specifically, we compare our MaD method (which assumes and fits features to a Gaussian distribution) against k-NN (k-nearest neighbors, a non-parametric method making no explicit distributional assumptions) and TWIN-Merging (which utilizes an MLP-based learned router to fit features). MaD and k-NN are both training-free, each utilizing only 64 samples per task based on Mahalanobis and cosine distances, respectively. By contrast, TWIN-Merging employs a **learned router** and requires a significantly more samples (1000 samples) per task for its task classification. For a more robust verification, these evaluations were performed across various ViT backbone models ({B/32, B/16, L/14}) and different numbers of tasks (8, 14, and 20 tasks).
>
> - Number of tasks = 8
> |  | **MaD** | k-NN (k=1) | k-NN (k=3) | k-NN (k=5) | k-NN (k=10) | TWIN |
> | --- | --- | --- | --- | --- | --- | --- |
> | ViT-B/32 | **98.0** | 93.7 | 92.5 | 91.7 | 89.4 | 90.5 |
> | ViT-B/16 | **98.3** | 94.7 | 93.6 | 92.8 | 91.3 | 86.8 |
> | ViT-L/14 | **98.5** | 93.2 | 92.8 | 92.2 | 90.7 | 83.7 |
> - Number of tasks = 14
> |  | **MaD** | k-NN (k=1) | k-NN (k=3) | k-NN (k=5) | k-NN (k=10) | TWIN |
> | --- | --- | --- | --- | --- | --- | --- |
> | ViT-B/32 | **96.9** | 92.9 | 92.0 | 91.2 | 89.1 | 84.0 |
> | ViT-B/16 | **97.0** | 93.4 | 92.4 | 91.7 | 90.2 | 79.9 |
> | ViT-L/14 | **97.7** | 93.7 | 93.5 | 93.2 | 92.0 | 72.6 |
> - Number of tasks = 20
> |  | **MaD** | k-NN (k=1) | k-NN (k=3) | k-NN (k=5) | k-NN (k=10) | TWIN |
> | --- | --- | --- | --- | --- | --- | --- |
> | ViT-B/32 | **94.1** | 90.0 | 89.5 | 89.0 | 86.8 | 75.8 |
> | ViT-B/16 | **94.7** | 90.4 | 89.4 | 88.8 | 87.0 | 77.6 |
> | ViT-L/14 | **95.8** | 91.3 | 91.2 | 91.3 | 90.2 | 78.8 |
>
> Our results demonstrate that **MaD consistently achieves strong task classification performance** across all diverse settings. This suggests that in scenarios where the true feature distribution is unknown, approximating it with a Gaussian distribution (as done in MaD) can be more effective than either making no explicit distributional assumption (k-NN) or using a learned router (TWIN-Merging). Particularly, we note that **MaD provides consistently better task classification performance than TWIN-Merging, even though TWIN-Merging employs a *learned router* with substantially more number of examples (i.e., 1000), while** **MaD is a *training-free approach with significantly less number of examples* *used for modeling Gaussian distribution* (i.e., 64)**.
>
> While acknowledging that assuming a Gaussian distribution may still not perfectly reflect the true underlying feature distribution, our experimental results indicate that **our Mahalanobis-distance-based task classification offers a highly effective and practical method when the actual distribution is unknown, compared to other approaches.** We agree that exploring even better alternatives to the Gaussian assumption is a promising avenue for future research.

---

> ### Author Response · Authors · 2025-11-26
>
> > Computational and numerical analysis of covariance estimation in very high-dimensional feature spaces.
>
> We thank the reviewer for raising the important point regarding the computational and numerical aspects of covariance matrix estimation in high-dimensional spaces. We have conducted further analysis to address these concerns.
>
> - Computational cost
> | Model  | ViT-B/32 | ViT-L/14 | T5-Large | LLaMA3-8B |
> | --- | --- | --- | --- | --- |
> | Dimension | 512 | 768 | 1024 | 4096 |
> | Time (per task) | 0.0023s | 0.0031s | 0.0043s | 0.0306s |
> | Normalized Performance (MaD) | 99.1 | 99.1 | 98.5 | 99.1 |
>
> Firstly, regarding the computational cost, we have analyzed the complexity of covariance calculation for increasing dimensions. Theoretically, the time complexity for computing the covariance matrix is $O(ND^2)$, and for its inverse, it is $O(D^3)$, where $N$ is the number of samples and $D$ is the dimension. Thus, the computational cost indeed increases cubically with dimension. However, it is crucial to note that this computation is performed **only once prior to inference and the resulting covariance matrix and its inverse are cached. Caching amortizes the cost and removes the need to compute covariance and its inverse for each sample during inference.** In practical terms, even for a very high dimension of $D=4096$, the pre-computation of the covariance matrix and its inverse takes only approximately 0.03 seconds per task, indicating that the actual computational time for the preparation of covariance matrix and its inverse is not prohibitively large.
>
> Secondly, to verify numerical stability as dimensions increase, we measured performance across models with varying feature dimensions: ViT-B/32 (D=512), ViT-L/14 (D=768), T5-Large (D=1024), and an extremely large LLaMa3-8B (D=4096). In all these cases, we consistently add a small, fixed valued epsilon to the diagonal elements of the covariance matrix for better stability, as described in our paper and mentioned by the reviewer. Our experimental results show that inference proceeded normally, and high normalized performance has been maintained across all tested dimensions. This indicates that by appropriately adding a small value to the diagonal elements of the covariance matrix, numerical instability issues due to high dimensionality (up to D=4096) can be effectively mitigated.

---

> ### Author Response · Authors · 2025-11-26
>
> > Generalizability to other model architectures (LLM).
>
> To verify the generalizability of our method across different model architectures, we have additionally validated on a large language model, specifically the LLaMA3.1-8B model.
>
> For this evaluation, we have used the following training tasks [1]: for the "mathematics reasoning" domain, we have used Tulu 3 Persona MATH, OpenMathInstruct 2, and MuminaMath-TIR; for the "general instruction following" domain, we have used WildChat, OpenAssistant, and No Robots; for the "knowledge recall" domain, we have used FLAN v2, SciRIFF, and TableGPT; and for the "precise instruction following" domain, we have used Tulu 3 Persona IF.
>
> Conversely, the test tasks have included GSM8K [2] and MATH [3] for the "mathematics reasoning" domain; BBH (CoT) [4] and DROP [5] for the "general instruction following" domain; PopQA [6] for the "knowledge recall" domain; and IFEval [7] for the "precise instruction following" domain. While the types of domains in the training and test sets have been the same five domains listed above, the specific tasks used within each domain have been configured differently. In other words, all tasks in the test set are unseen.
>
> | Model/Domain | Math | General IF | Precise IF | Knowledge | Avg |
> | --- | --- | --- | --- | --- | --- |
> | Fine-tuned | 0.528 | 0.650 | 0.715 | 0.295 | 0.547 |
> | Weight Averaging | 0.467 | 0.663 | 0.577 | 0.319 | 0.507 |
> | Task Arithmetic | 0.468 | 0.662 | 0.584 | 0.319 | 0.508 |
> | TIES-Merging | 0.528 | 0.655 | 0.610 | 0.313 | 0.527 |
> | EMR + MaD (Ours) | 0.552 | 0.640 | 0.656 | 0.318 | 0.542 |
> | TM-TA + MaD (Ours) | 0.546 | 0.636 | 0.608 | 0.318 | 0.527 |
>
> **The experimental results show that our method outperforms other merging baselines in “mathematics reasoning” and “precise instruction following”.** Notably, in “mathematics reasoning”, our approach surpasses not only the merging baselines but also the performance of the single fine-tuned model. Although our score in “general instruction following” and “knowledge recall” is marginally lower than the top-performing Weight Averaging method, the difference is very small. **This demonstrates that our method works well for LLMs and exhibits outstanding generalization capability, even on unseen tasks.**
>
> [1] Lambert et al., Tulu 3: Pushing Frontiers in Open Language Model Post-Training, arXiv, 2024.
>
> [2] Cobbe et al., Training Verifiers to Solve Math Word Problems, arXiv, 2021.
>
> [3] Hendrycks et al., Measuring Mathematical Problem Solving With the MATH Dataset, NeurIPS, 2021.
>
> [4] Suzgun et al., Challenging BIG-Bench Tasks and Whether Chain-of-Thought Can Solve Them., Findings of ACL, 2023.
>
> [5] Dua et al., DROP: A Reading Comprehension Benchmark Requiring Discrete Reasoning Over Paragraphs, NAACL, 2019.
>
> [6] Mallen et al., When Not to Trust Language Models: Investigating Effectiveness of Parametric and Non-Parametric Memories, ACL, 2023.
>
> [7] Zhou et al., Instruction-Following Evaluation for Large Language Models, arXiv, 2023.

---

### Author Response · Authors · 2025-11-26
**General response**

We sincerely thank the reviewers for their time and thoughtful reviews. The insightful questions and helpful feedback received have been valuable, greatly improving the quality of our work.

Below, we list the main strengths of our paper that were identified by reviewers. Then, we address the common points raised in the comments, sharing the additional experiments we have done.
We hope this summary and our detailed answers to each reviewer help clear up any concerns. We look forward to discussing further and providing any more information the reviewers may request during the discussion period.

## **Highlighted Strengths of our work**

We are encouraged that reviewers recognized our work to

- be novel and well-motivated  (**`2yxj`, `Jwgd`**)
- be simple yet effective plug-and-play solution (**`2yxj`, `Jwgd`**)
- be well-supported by extensive experiments (**`2yxj`**)
- demonstrate clear performance improvements over strong baselines like TWIN-Merging and DaWin (**`2yxj`, `75xv`, `Jwgd`**)
- be well-written, logically structured, and well-designed (**`2yxj`, `75xv`, `5sgK`**)
- be efficient compared to existing approaches (**`5sgK`**)
- address a critical real-world problem (tasks unknown at inference) and a major barrier to deploying models in real-world systems (**`5sgK`, `Jwgd`**)

## **Main contributions of our work**

- **Efficient and Training-Free Task Classification for Dynamic Merging**

    We propose MaD, a **training-free approach** that enables dynamic model merging for task-unknown scenarios with **significantly fewer examples used for modeling Gaussian distributions** (our task classification). MaD not only drastically reduces memory cost (by over 5 times compared to existing dynamic merging approaches, requiring storage of only merged task vectors and task-specific masks instead of full models), but also facilitates inference with a single forward pass per sample, thus avoiding the linear increase in task-specific model forward passes typically found in other dynamic methods.

- **Empowering Existing Subspace-based Merging for Task-Unknown Scenarios**

    MaD offers a plug-and-play solution that allows existing subspace-based model merging methods to effectively operate in task-unknown scenarios. By reformulating this challenge as a task classification problem under Gaussian Discriminant Analysis, MaD extends and enhances the performance of previous merging techniques under more realistic settings, where task information is unavailable.

- **Effective and Efficient Across Diverse Tasks (Vision & NLP) and Networks (including LLMs)**

    Experimental results demonstrate MaD's effectiveness and flexibility across both computer vision and natural language processing domains under task-unknown scenarios.

## **Additional Experiments**

- **Additional Baselines for Multi-task Performance and Inference Cost (`75xv`, `5sgK`):**

    We have conducted further comparisons against prominent dynamic merging approaches, including WEMoE and MoW-Merging in the Table 2 (Line 324 - 347) of revision. Our results consistently demonstrated that MaD achieved better **performance and lower inference cost** when benchmarked against these methods in the Table 4 (Line 432 - 443) of revision.

- **Robustness to Out-of-Distribution (OOD) Scenarios (`5sgK`) :**

    We evaluated MaD's performance on unseen tasks in Table 5 (Line 445 - 458) of revision. Our findings showed that MaD exhibited **stronger performance on unseen tasks** compared to other methods, and demonstrated **robustness even under feature distribution drift.**

- **Scalability to Larger Task Sets (`5sgK`):**


    To assess scalability, we extended our evaluation to 30 and 50 computer vision tasks in Table F4 (Line 1188 - 1196) and Table F5 (Line 1197 - 1217) of revision. MaD consistently showed **strong task classification performance** and achieves **the best multitask performance** even with this increased number of tasks.

- **Extension to LLM (`2yxj`, `75xv`)**

    We further validated MaD's efficacy and generalization capabilities by testing it on the LLaMA3.1-8B model in Table F8 (Line 1368 - 1382) of revision. This confirms the robustness of MaD when applied to LLMs, showcasing its excellent ability to generalize even to tasks it has not encountered.

---

### Comment · Area_Chair_w4sr · 2025-11-27
**Request for Timely Response to Authors’ Rebuttal and Discussion**

Dear Reviewers,

I hope you are doing well. The authors have now submitted their rebuttal for the paper under your review. At this stage, your timely response is essential for ensuring a smooth discussion phase.

Could you please review the rebuttal at your earliest convenience and share your updated thoughts? If there are points that require further discussion among the reviewers, please feel free to initiate or join the conversation on the discussion thread.

Your prompt input will greatly help us maintain the review timeline. Thank you very much for your efforts and valuable contributions.

Best regards,

AC

---

### Author Response · Authors · 2025-12-03
**Summary for Area Chair**

We thank the Area Chairs and Program Chairs for their careful consideration of our submission. We appreciate the time and hard work invested to keep the conference working, especially during this busy period in the amidst the unforeseen issues.

To assist the final evaluation, a brief overview of the main contributions of our work and the consensus from the rebuttal is provided below.

## **Highlighted Strengths of our work**

- MaD is a **novel, training-free classification solution for task-unknown model merging**, offering a **simple, plug-and-play alternative** that avoids costly training and multiple forward passes. (**`2yxj`, `5sgK`, `Jwgd`**)
- Well-supported by extensive experimental, which **consistently demonstrates clear performance improvements** over strong baselines such as TWIN-Merging and DaWin.  (**`2yxj`, `Jwgd`**)
- Our work **addresses the realistic and critical problem of task-unknown inference**, a significant deployment barrier that is currently under-addressed in merging literature. (**`5sgK`, `Jwgd`**)

## **Main Concerns Addressed**

> **Additional Baselines for Multi-task Performance and Inference Cost**
>

Additional evaluations against dynamic merging methods (WEMoE [1], MoW-Merging [2]) show that MaD achieves **higher accuracy while reducing inference cost**. (in Section 5.2 and 5.3 of revision)

> **Robustness to Out-of-Distribution (OOD) Scenarios**
>

We have evaluated MaD on **6 unseen tasks**, and it outperforms task-unknown baselines (TWIN-Merging [3], DaWin [4]), showing **strong robustness to distribution shifts**. (in Section 5.2 of revision)

> **Scalability to Larger Task Sets**
>

New experiments with **30- and 50-task** CV benchmarks show that MaD **maintains strong task classification accuracy and achieves the best multitask performance**. (in Appendix F: Scalability to Larger Task Sets of revision)

> **Extension to LLM**
>

We have evaluated our method on **LLaMA3.1-8B**, where MaD shows **outstanding average performance** in comparison to other model merging methods. (in Appendix F: Performance on a large language model of revision)

> **Extended analysis of feature Gaussian distribution**
>

Comparing MaD with k-NN and a trained router (TWIN-Merging [3]) shows that MaD **achieves better task-classification accuracy with only 64 samples**, supporting the practicality of the Gaussian assumption. (in Appendix F: Task classification performance of revision)

Beyond these main points, we have also carefully **addressed** **all other concerns raised by the reviewers**, making corresponding revisions throughout our work.

[1] Tang et al., Merging Multi-Task Models via Weight-Ensembling Mixture of Experts, ICML, 2024.

[2] Ye et al., Dynamic model merging with mixture of weights, IEEE Transactions on Circuits and Systems for Video Technology, 2025.

[3] Lu et al., Twin-Merging: Dynamic Integration of Modular Expertise in Model Merging, NeurIPS, 2024.

[4] Oh et al. Dawin: Training-free dynamic weight interpolation for robust adaptation, ICLR, 2025

---

### Meta-Review · Area_Chair_JARS · 2025-12-17

**Summary:**

The paper introduces MaD, a training-free framework designed to allow subspace-based model merging methods to operate in "task-unknown" scenarios in which task identity is unknown during inference. MaD models the core challenge as a task classification problem and uses Gaussian Discriminant Analysis to model the feature distribution of each task using features extracted from a merged model. At inference time, MaD calculates the Mahalanobis distance between the input and each task distribution to select the task corresponding to the minimum distance. Experimental results are given on a range of merging benchmarks which demonstrate the effectiveness of augmenting various task-arithmetic based aggregation approaches with MaD.

The main strength of the proposed approach is its "plug-and-play" nature, which allows it to be applied to virtually any merged model (it is, in the end, just a task classifier). This main advantage, however, is also its main weakness: it is, in the end, *just a task classifier*.

Reviewers voiced a number of concerns which affected the final recommendation:
+ A major concern was the assumption that features from the merged model follow a multivariate Gaussian distribution for each task, which is a strong assumption with little theoretical motivation.
+ Reviewers also suggested the proposed method was straightforward, with limited technical contribution, and that it leverages the good task separation capabilities of underlying merged models more than it proposes a novel approach.
+ Reviewers noted a conceptual tension because the task classification relies on features extracted from a model created via simple weight averaging, a method which generally performs poorly on the end multi-task predictions.
+ Reviewers initially noted a lack of comparison with conceptually similar dynamic merging approaches (such as WEMoE and MoW-Merging)
+ Finally, one reviewer noted numerous typographical, grammatical, and symbolic errors in the initial submission.

**Reviewer Concerns:**

Reviewer concerns adequately addresses:
+ Concerns raised regarding efficiency due to MaD requiring an initial forward pass for feature extraction before final inference, which introduces additional inference-time overhead, were adequately addressed in rebuttal.
+ Similarly, the data requirement concerns, since MaD relies on extracting the mean and covariance of the data, were adequately addressed in rebuttal with the authors pointing out that most task-agnostic approaches require *some* parameter estimation using original task data.
+ The question of scalability was sufficiently addressed in rebuttal, with additional experiments on longer task sequences.
+ Related to the previous point, new rebuttal experiments on fine-grained datasets adequately addressed concerns regarding feature drift.

Outstanding reviewer concerns:
+ **Novelty and Significance**: The core contribution of the paper is a nearest-task classifier based on the Mahalanobis distance. As such, the overall technical contribution of the work is extremely limited. Without significant additional theoretical analyses and motivations, it does not seem to be a paper with broad appeal to the community.
+ **Gaussian Assumption**: While the authors provided additional *empirical* results comparing Mahalanobis distance task classification to nearest-neighbor and learned task classifiers, the question of the appropriateness of the Gaussian assumption from a theoretical motivation point of view remains open.
+ **Comparison with Dynamic Merging**: Although partially addressed in rebuttal, the work requires an experimental evaluation much more concentrated on comparing with approaches with similar aims (test-time adaptation to the task at hand).

**Reviewer Scores:**

+ **R1 (2yxj)**: A somewhat shallow initial review, I do not expect this reviewer would have significantly changed their score or engaged significantly in the discussion.

+ **R2 (75xv)**: The most detailed and articulated review. I expect this reviewer would have conceded points related to additional data requirements, and perhaps even on dynamic merging during the discussion period. However, I do not believe they would have been convinced that the contributions of the paper were significantly novel based on the new empirical results in rebuttal.

+ **R3 (5sgK)**: A middle-of-the-road review. This reviewer might have been convinced by the scaling and feature drift experiments, but I do not see them significantly raising their.

+ **R4 (Jwgd)**: Another somewhat weak initial review, making it hard to gauge how this reviewer would have engaged in the discussion process. It seems unlikely they would have increased their score significantly.

In summary, there was no initial enthusiasm of note towards accepting this paper, and it seems unlikely -- given the limitations in terms of novelty and significnace of the contribution -- that there would have been a strong consensus towards accepting in the end.

---

### Decision · Program_Chairs · 2026-01-26

Reject